# Green moisture-electric generator based on supramolecular hydrogel with tens of milliamp electricity toward practical applications

Su Yang [1,2], Lei Zhang[3], Jianfeng Mao[4], Jianmiao Guo [4], Yang Chai [4], Jianhua Hao [4], Wei Chen [5] & Xiaoming Tao [1,2] ✉

Moisture-electric generators (MEGs) has emerged as promising green technology to achieve carbon neutrality in next-generation energy suppliers, especially combined with ecofriendly materials. Hitherto, challenges remain for MEGs as direct power source in practical applications due to low and intermittent electric output. Here we design a green MEG with high direct-current electricity by introducing polyvinyl alcohol-sodium alginate-based supramolecular hydrogel as active material. A single unit can generate an improved power density of ca. 0.11 mW cm$^{-2}$, a milliamp-scale short-circuit current density of ca. 1.31 mA cm$^{-2}$ and an open-circuit voltage of ca. 1.30 V. Such excellent electricity is mainly attributed to enhanced moisture absorption and remained water gradient to initiate ample ions transport within hydrogel by theoretical calculation and experiments. Notably, an enlarged current of ca. 65 mA is achieved by a parallel-integrated MEG bank. The scalable MEGs can directly power many commercial electronics in real-life scenarios, such as charging smart watch, illuminating a household bulb, driving a digital clock for one month. This work provides new insight into constructing green, high-performance and scalable energy source for Internet-of-Things and wearable applications.

Low-carbon technologies play a critical role in global quest for carbon neutrality, which can be achieved by using clean and reusable energy. The emerging moisture-motivated energy is a promising green alternative as a next-generation power source for broad internet-of-things (IoTs) and wearable applications[1,2]. Owing to the ubiquity of atmospheric moisture, moisture-electric generator (MEG) is superior in terms of inexhaustibility, sustainability, and convenience beyond the environmental and regional restrictions. Since the pioneering work by

Qu et al. in 2015[3,4], those years have witnessed the MEG field is flourishing vigorously with increasing electric output by new materials and structure design. In the early stage, intermittent electric energy was generated by MEGs made from graphene oxide film, graphene derivatives and titanium dioxide by virtue of water gradient[5–7]. Later on, spontaneously sustained electricity has been demonstrated in hydrophilic polymer membranes with rich ionic groups, as well as bacterial protein nanowires with nanoporous structure[8–10]. Beyond that, the

[1]Research Institute for Intelligent Wearable Systems, The Hong Kong Polytechnic University, Hong Kong, P. R. China. [2]School of Fashion and Textiles, The Hong Kong Polytechnic University, Hong Kong, P. R. China. [3]Department of Materials Science and Engineering, University of Science and Technology of China, Hefei, P. R. China. [4]Department of Applied Physics, The Hong Kong Polytechnic University, Hong Kong, P. R. China. [5]National & Local Joint Engineering Research Center for Textile Fiber Materials and Processing Technology, School of Materials Science and Engineering, Zhejiang Sci-Tech University, Hangzhou, P. R. China. ✉e-mail: xiao-ming.tao@polyu.edu.hk

output voltage has jumped several orders of magnitude, from a few millivolts by a single MEG unit to several hundreds of volts by large-scale integrated MEG bank[11,12]. Despite the fruitful advance of these pioneering research, progress in the output current thus power is rather lagged behind. Majority of MEGs were reported with a low current density of below 10 μA cm$^{-2}$ and resultant power density of below 10 μW cm$^{-2}$, making it difficult to scale up to a practical level[13]. Hitherto, present MEGs need to depend on auxiliary energy-storage devices and rectification circuits to achieve small electronics driving[13], which may bring in additional energy loss and increased complexity in circuit design. Hence, it is highly desirable but challenging to develop a direct-current (DC) MEG with high power output capable of directly driving IoTs devices.

In fact, the power output of state-of-arts MEGs is mainly decided by water capturing ability and constructed water gradient[14,15]. MEGs rely on the interaction of active materials with moisture[16]. The improved water capturing capability can empower MEG with sufficient chemical conversion energy to trigger ample ions diffusion, consequently inducing high power output. Hence, the moisture-initiated ion diffusion is considered as the driving force for electric generation. Also, the remained water gradient guarantees a continuous electric output. However, excess water intake may cause quick equilibrium at both ends, leading to decayed electricity output[17]. Thereby, it is difficult to achieve great water capturing and long-term maintained water gradient at the same time. Great efforts have been made to enhance moisture absorption capability by the physical-chemical modifications to improve surface hydrophilicity[7,18] and creating hierarchical pores to increase specific contact area between active materials and water molecules[3,19]. Additionally, durable water gradient relies on not only built-in chemical gradient by thermal reduction, laser irradiation, polarization, or heterogeneous composites[9,20,21], but also providing asymmetric moisturization by unidirectional moisture stimuli or asymmetric humidity environment[4]. The feasible strategies are to combine above methods to achieve high moisture absorption with a sustained moisture gradient. For example, heterogeneous composites with different wettability can promote self-sustained electric generation by the efficient absorption, transmission, and evaporation of water[20]. Also, materials with high porosity and ion density gradient not only improve the contact area but also construct remained water gradient, leading to an enhanced electricity[22]. Besides the high power output, developing green MEG is significant to find fully clean, low-carbon and sustainable energy alternatives. It requires not only sustained energy source and conversion process, but also the adoption of environmentally friendly materials and fabrication process. However, previous works mainly focus on the power output performance, while the used materials, like electrospun nanofiber[23] and partially-reduced graphene oxide[21], often involve complex, power-consuming and costly fabrication process. Thus, it is meaningful to explore green materials with facile fabrication method.

To address the above issues, here we elaborately designed a green MEG with DC power output by introducing polyvinyl alcohol (PVA)−sodium alginate (AlgNa)-based supramolecular hydrogel as active material. Both of PVA and AlgNa have the merits of biocompatibility, biodegradability, easy processing, and low cost[24]. And the preparation process is simple, efficient and pollution-free. Especially, the supramolecular hydrogel is expected to feature "fast absorption" of water and "relatively slow diffusion" of ion water clusters. "Fast absorption" origins from a wealth of water affinity sites of AlgNa to enhance water uptake, helpful for ion dissociation and diffusion. "Relatively slow diffusion" enables trapping water nearby to maintain water gradient through asymmetric absorption of moisture, beneficial to a sustained electric output. Beyond expectation, a single MEG unit can produce a high open-circuit voltage ($V_{oc}$) of ca. 1.30 V, a sustained DC current of 2.14 mA with 9 cm$^2$ size, as well as an excellent peak power density of 0.11 mW cm$^{-2}$ with 1 cm$^2$ size, which transcends

most of the counterparts. More importantly, a surpassing enlarged current of ca. 65 mA has been achieved by high-efficient parallel connection. The scalable and continuous electric output can directly power many commercial electronic devices without the need of extra rectifying circuits and energy storage devices, such as charging smart watch, lighting a household bulb illumination, driving a digital clock for 1 month. Manifested by experiments and density functional theory (DFT) calculations, such high performance of MEG origins from the synergistic effects of enhanced moisture absorption ability, long-term water gradient, and ample dissociable ions with improved ionic conductivity within PVA-AlgNa-based hydrogel. The exceeding electric performance as well as practical applications carry profound implications for designing future MEG device, spurring the development of MEGs as a green and sustainable power source toward carbon neutrality.

## Results
### Structure and electric output of one MEG device
As shown in Fig. 1a, one single MEG device consists of a green electricity-generating layer and a pair of asymmetric electrodes. The electricity-generating layer is well-designed by molecular engineering AlgNa into PVA hydrogel. Non-ionic polyhydroxy PVA is easy to construct a physical cross-linked network with hydrophilic property. As a natural polysaccharide, the polyanionic AlgNa features with numerous hydroxyl groups (Supplementary Fig. 1), thus presenting prominent water-affinity feature, which is expected to enhance the water absorption of MEG. Moreover, by adding crosslinker of CaCl$_2$, an obvious blue shift of −COO− stretching band is observed in Fourier Transform Infrared Spectroscopy (FTIR) spectrum, suggesting the crosslinking Ca$^{2+}$ with −COO− of AlgNa[25]. That means supramolecular AlgCa/Na ionically cross-linked network is formed with abundant carboxyl functional groups (e.g., -COONa and -COOCa) as depicted in Fig. 1a[26], which plays a key role as dissociable ions when interacting with water molecules. Besides, a pair of asymmetric electrodes were established by directly laser-induced graphene as bottom electrode on polyimide (PI) substrate and aluminum (Al) film with holes as top electrode, which are conducive to large-scale integration efficiently and improve output performance of MEG device[10,17]. Additionally, Al film is highly flexible, lightweight, easily accessible and fairly cheap, which is desirable for scalable and low-cost MEGs towards wide applications[27,28]. Poly(3,4-ethylenedioxythiophene): poly(styrene sulfonate) (PEDOT:PSS) was pre-coated on bottom electrode as the mediation of electrodes/materials interfaces and carrier transport-assisting layer[29], thus presenting an enhanced voltage and current output as shown in Supplementary Fig. 2. The moisture in air goes through porous top electrode but blocked by bottom PI substrate, spontaneously forming water gradient firstly.

The enhanced water absorption capability is identified by the water affinities of simplified PVA, AlgNa, AlgCa polymers via DFT calculation of the molecular electrostatic potential (ESP) distribution on the molecular van der Waals (vdW) surface[30,31]. ESP maps in Fig. 1b illustrate the three-dimensional charge distribution of PVA, AlgNa, AlgCa. The minimum and maximum ESP values of PVA are only −33.72 and 40.53 kcal mol$^{-1}$. The ESP distribution of PVA shows inconspicuous discrepancy with large area (Supplementary Table 1) overwhelmed by green region, which indicates a relatively weak moisture affinity[32]. In clear contrast, AlgNa and AlgCa show high ESP charge distribution near Na and Ca elements, corresponding to the maximum ESP values of 133.50 and 184.58 kcal mol$^{-1}$, respectively. Importantly, AlgNa and AlgCa in Fig. 1c exhibit vast vdW surface area proportion (ca. 20% and 30%, respectively) featuring high absolute ESP values (i.e. <−40 kcal mol$^{-1}$ and >+40 kcal mol$^{-1}$) (Supplementary Tables 2 and 3). The greater area with larger absolute value implies better moisture absorption capability in air[33,34]. The result indicates that the AlgCa/Na improves moisture capturing capability of MEG, in favor of high power

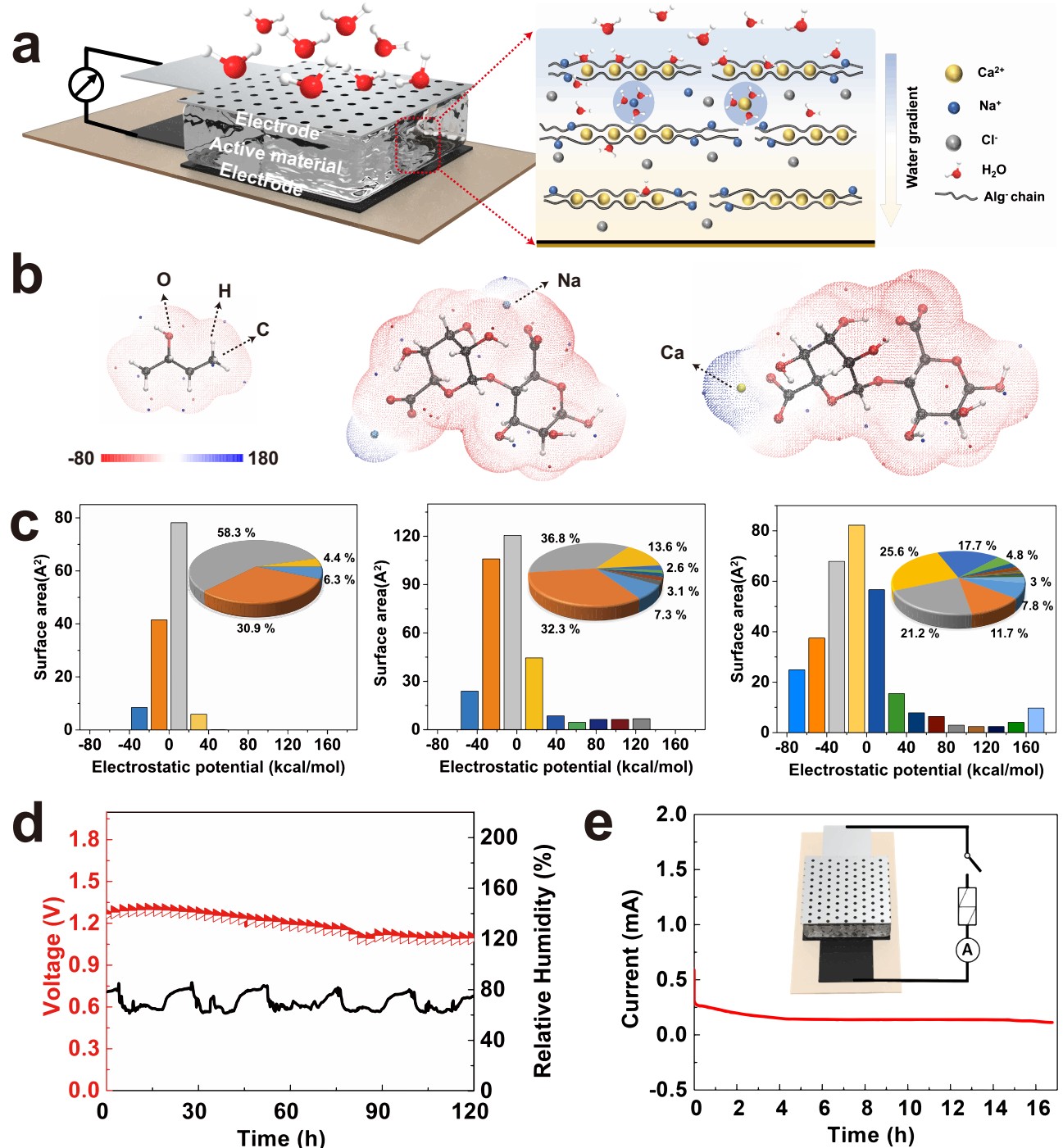

**Fig. 1 | Device structure of moisture-electric generator (MEG) and corresponding electric output. a** Schematic illustration of a single MEG unit. The right scheme shows that active material consists of supramolecular AlgCa/Na network within PVA hydrogel, which is formed through replacing partial $Na^+$ ions by $Ca^{2+}$ ions as the egg-box crosslinked points. Water gradient is defined as the difference in moisture content between upper side and lower side of the MEG sample. **b** The electrostatic potential (ESP) distribution of PVA, AlgNa, and AlgCa from left to right by DFT calculations. The units of color bar are in kcal mol$^{-1}$. Surface local minima and maxima of ESP are represented as orange and blue spheres, respectively. **c** Surface area and corresponding area percent in each ESP range on the vdW surface of PVA, AlgNa, AlgCa from left to right correspondingly. **d** The $V_{oc}$ (red curve) of an MEG device over time under an open environment with fluctuating RH (relative humidity). The ambient RH (black curve) was synchronously recorded. **e** The current output of an MEG device with 1 kΩ external resistor under the evolution of time at 70% RH. The insert shows the circuit diagram of MEG with one resistor. The area of one single unit is fixed at 1 cm² and the test temperature is about 22 °C, unless otherwise stated. Source data are provided with this paper.

output. Beyond expectation, a single MEG unit as shown in Fig. 1d generates a continuous DC $V_{oc}$ of about 1.30 V for more than 90 h in an open ambient environment with relative humidity fluctuating from 60% to 90% and temperature of ca. 22 °C, reflecting its excellent stability over a long term. More significantly, a MEG unit generates a large

current output of ca. 0.25 mA when connected to an external resistance of 1 kΩ and keeps continuous output for about 17 h (Fig. 1e). Beyond that, the current output gradually decays due to saturation of moisture absorption but still persists for over 1 month (Supplementary Fig. 3). Such a DC MEG with sustained power output is highly desirable,

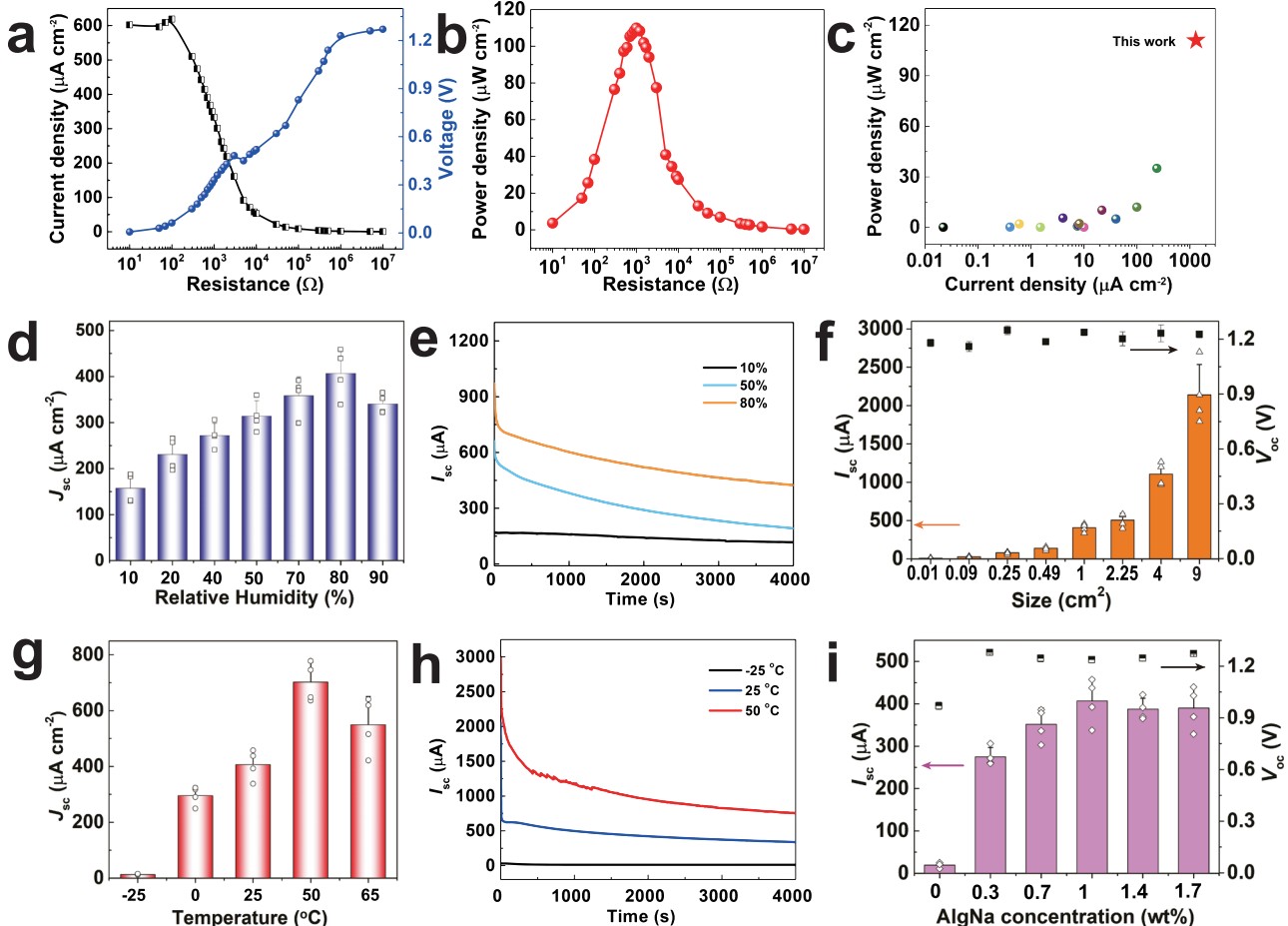

**Fig. 2 | The output performance of one moisture-electric generator (MEG) unit.**
**a** The electric output of MEG unit with external resistance varied from $10^1$ to $10^7$ Ω at 85% RH. **b** The dependence of power density on external resistance according to (**a**). **c** The performance comparison of reported sustained MEGs with DC output under only moisture stimulation. The active materials used as shown in (**c**) follow the order from left to right: protein nanofiber, silicon nanowires, asymmetric GO, PSS/PVA textile, PDDA-PSS-PVA, LiCl-loaded cellulon paper, AAO, polydiallyl dimethylammonium chloride, GO-PAAS, PNIPAM, protein nanowire, SA/SiO$_2$/RGO, PVA-PA-Glycerol, PVA-AlgNa-based hydrogel (this work)[8–10,14,20,21,36,43,51–55]. **d** $J_{sc}$ of MEG under different RH (10–90%). Data represent the mean ± standard deviation ($n = 4$). **e** The $I_{sc}$ curves with the time under 10%, 50%, 80% RH. **f** The $I_{sc}$ and $V_{oc}$ of MEG at different sizes from 0.01 to 9 cm$^2$ at 80% RH. Data represent the mean ± standard deviation ($n = 4$). **g** $J_{sc}$ of MEG under different temperatures from −25 to 65 °C. Data represent the mean ± standard deviation ($n = 4$). **h** The $I_{sc}$ curves with the time at temperature of −25, 25, 50 °C. **i** The $I_{sc}$ and $V_{oc}$ of MEG at different AlgNa concentration from 0 to 1.7 wt% at 80% RH. Data represent the mean ± standard deviation ($n = 4$). Source data are provided with this paper.

which achieves the top level among existing sustained MEGs as discussed in Fig. 2c hereinafter.

### Electric output performance of one MEG unit

The electric output performance of our newly developed MEG is investigated comprehensively at different conditions. Firstly, connecting with external resistance from 10 Ω to 10 MΩ, the output voltage increases while the current density decreases (Fig. 2a). A maximum power density of 0.11 mW cm$^{-2}$ is achieved at an optimal resistance of 1 kΩ for one single MEG unit with only 1 cm$^2$ area (Fig. 2b). Notably, our MEG unit demonstrates superior power density and current density, compared to most of sustained MEGs under only moisture stimulation (Fig. 2c). Indeed, majority of MEGs suffer from very low current density and ungratified power density (orange part in Fig. 2c). On the contrary, our single MEG unit can deliver an exceeding short-circuit current density ($J_{sc}$) of 1.31 mA cm$^{-2}$ and a predominant power density of 0.11 mW cm$^{-2}$, promising a direct boost of MEGs to drive electronic devices for practical use.

The humidity-dependent characteristics of MEGs are measured under wide range of RHs (Fig. 2d). With the increase of RH from 10 to 80%, the average $J_{sc}$ augments monotonically up to 407 μA cm$^{-2}$, well

corresponding to enhanced moisture uptake capability of MEG (Supplementary Fig. 4a). The escalated $J_{sc}$ and moisture uptake capability hint the essential role of moisture in power generation. A slight decrease of $J_{sc}$ is observed at 90% RH possibly due to reduced water gradient with saturated absorption. Figure 2e shows the detailed current curves of MEGs at different RH conditions. A continuous DC current of about 450 μA is maintained even beyond 4000 s at 80% RH. It is notable that a favorable $J_{sc}$ (ca. 150 μA cm$^{-2}$) is observed at 10% RH, derived from strong hygroscopicity of MEG to trigger ion diffusion even at low RH (Supplementary Fig. 4b). Besides, $V_{oc}$ presents a gradual rise from ca. 0.8 to 1.3 V with RH rising to 80% (Supplementary Fig. 5a), due to a larger ion concentration difference formed at an improved RH. MEGs dried at suitable condition show similar electric outputs after exposed in the same test environment (Supplementary Fig. 6), indicating that the strong moisture absorption capability is critical to electric generation. In addition, our MEG also shows great adaptability under wide range of temperature. From −25 to 50 °C, the average $J_{sc}$ shows a sharp climb from 13 to 703 μA cm$^{-2}$. Such obvious growth benefits from the synergistic effect of accelerated ion transport rate and ion concentration at high temperature. While at 65 °C, the slightly decreased current density probably results from restricted ion

diffusion by massive water desorption from MEG. Figure 2h clearly exhibits a large DC current curve output of about 756 μA with the time evoluting into 4000 s at 50 °C. The elevated temperature also improves $V_{oc}$ from 0.9 V slowly until reaching a platform of ca. 1.3 V (Supplementary Fig. 5b), mainly due to enhanced ion concentration difference at high temperature. Furthermore, MEGs with different top electrodes output favorable electric performance and display similar humidity/temperature-dependent characteristics in Supplementary Fig. 7, suggesting that electricity generation mainly derives from moisture absorption by supramolecular hydrogel. The comparable $V_{oc}$ and $I_{sc}$ outputs are also found in an oxygen-insulated environment under 80% RH, further excluding the influence of electrode (Supplementary Fig. 8)[35]. Besides, the change of relative electrode positions exerts limited influence on electricity performance of MEG (Supplementary Fig. 9). The all-weather adaptability and great electric output performance enable the newly developed MEG device to work as versatile power source in most of the environmental conditions.

Furthermore, the device area of MEG exerts remarkable effect on the current output. As shown in Fig. 2f, a MEG unit with only 0.01 cm² can generate a short-circuit current ($I_{sc}$) of about 13 μA, endowing an exceeding $J_{sc}$ of 1.31 mA cm⁻². Excitingly, $I_{sc}$ demonstrates two orders of magnitude upgrade (ca. 2.14 mA) when the size increases to 9 cm². The corresponding $I_{sc}$ curves for different sizes have been plotted in Supplementary Fig. 10a. Both MEGs with sizes of 4 and 9 cm² show large DC current outputs of more than 1 mA even beyond 4000 s. Meanwhile, the $V_{oc}$ remains almost the same despite the area changes, which is consistent with previous works[36]. One single MEG unit is able to generate both high $V_{oc}$ of ca. 1.30 V and sufficient $I_{sc}$ of ca. 2.14 mA under only ambient stimulation (22 °C, 80% RH), which considerably broadens the application scenarios of MEGs. Moreover, AlgNa and cross-linker CaCl₂ also affect the electric performance of MEG. As shown in Fig. 2i, both $V_{oc}$ and $I_{sc}$ are inferior without AlgNa, due to scarce ions and inferior water absorption capability. The $I_{sc}$ rises up to ca. 200 μA with the addition of only 0.3 wt% AlgNa. That means AlgNa plays a pivotal role on determining the current output of MEG. The $I_{sc}$ enhances gradually then keeps almost constant despite further increasing AlgNa concentration, which probably arises from the compromise between rigid AlgNa chains and increased dissociable ions source. The rigid AlgNa chains may retard ion diffusion and extend tortuous diffusion paths, imposing a side effect on $I_{sc}$. Besides, the incorporation of CaCl₂ brings in a rise of $I_{sc}$ firstly while a certain decline beyond 3.8 wt% of CaCl₂ (Supplementary Fig. 10b). The increased $I_{sc}$ stems from more moisture capturing and conducive ion diffusion within supramolecular network. However, the excessive CaCl₂ may lead to attenuated water gradient by absorbing excess water, thus lowering power output. Similar phenomenon is observed for MEGs with different thickness. MEG with 2.5 mm thickness shows decreased current density due to long ion diffusion path (Supplementary Fig. 10c). Here, the samples with AlgNa concentration of 1 wt %, CaCl₂ concentration of 2.4 wt% and 2 mm thickness are used in this work unless otherwise stated.

## Working mechanism of MEG

As we know, the improved water capturing capability can empower MEG with sufficient chemical conversion energy to trigger plentiful ions diffusion. The increased RH can enhance water absorption capability (Supplementary Fig. 4). The current in Fig. 3a represents a positive boost after changing RH from 5% to 75%. With the intermittent and periodic RH variation from 30% to 60%, a regular change in current output of MEG synchronously responses (Supplementary Fig. 11). In addition, a decent cyclic current output was also observed by moisture absorption-dehydration process (Supplementary Fig. 12a). These results further indicate that moisture is the prime energy source of MEG. Besides external humidity stimuli, the elevated output power is believed to benefit from plentiful dissociable ions of MEG. As displayed

in Fig. 3b, the non-ionic PVA delivers a very low current of ca. 0.14 μA, whereas PVA-AlgNa-based supramolecular hydrogel generates a much higher current of ca. 300 μA at the same condition. This is consistent with distinct moisture uptake capability in Fig. 3c, where the weight of PVA-AlgNa-based supramolecular hydrogel increases by 55 wt% after exposing in the air of over 250 min, while that of PVA remains almost unchanged. The several fold enhanced moisture intake capability not only enables MEG with plentiful ions dissociation but also triggers ion diffusion as well as enhances its ionic conductivity as shown in Supplementary Fig. 13, thus greatly escalating the generation of current output. A sustained current lasts for more than 120 h accompanied by the concurrent adsorption of water molecules, which couples with the ion dissociation process in the MEG even after reaching absorption saturation state (Supplementary Fig. 14). It well demonstrates that moisture absorption process is directly related to electricity generation of MEG.

Interaction region indicator (IRI) by DFT calculation further verifies the key role of the interaction between polymers and moisture on the output performance of MEG[37]. Figure 3d–g show the IRI isosurfaces of sign($\lambda_2$)$\rho$ between H₂O and different polymers, the calculated adsorption energies as well as the corresponding chemical explanations[38,39]. As shown in Fig. 3g, the estimated adsorption energy ($\Delta E$) of H₂O with AlgCa is −142.94 kJ mol⁻¹, and H₂O with AlgNa of −106.16 kJ mol⁻¹, implying a notable attraction between AlgCa/Na and H₂O. In stark contrast, a plunge (only −22.19 kJ mol⁻¹) happens on PVA and H₂O, which suggests a much weaker interaction. The weak interaction is also observed between Alginate acid and H₂O (−63.31 kJ mol⁻¹) (Supplementary Fig. 21c). The attractive force between H₂O and polymers can be further revealed by blue spikes in IRI scatter plots and corresponding blue regions in IRI maps. Sign($\lambda_2$)$\rho$ function mapped on IRI isosurfaces by different colors offers direct insight into nature of the interaction regions[37]. In Fig. 3d–f, it clearly reveals that the interaction strength between H₂O and polymers decreases with the order: AlgCa > AlgNa > PVA, which is well verified by the sustained durability of electrical outputs in Supplementary Fig. 15. In fact, these weak interactions (like vdW interaction) of the green regions in Fig. 3g(iii) prefer to exist between H₂O and PVA. Especially, the blue regions on the IRI maps between H₂O and AlgCa/Na (Fig. 3g(i, ii)) suggest notable attractive interaction, which is the key driving force for moisture adsorption, dissociation and trapping. Thus PVA-AlgNa-based hydrogel features "fast absorption" and "relatively slow diffusion" of water clusters conceivably with large adsorption energy and favorable hydrogel network. In other words, the supramolecular hydrogel improves water adsorption meanwhile maintains water gradient by trapping water nearby the interface with asymmetric moisture absorption.

Except for enhanced moisture uptake capability, the high electric generation of MEG also depends on remained water gradient, which is further verified by in situ Raman mapping. As depicted in Fig. 4a, in situ 2D Raman measurements were performed to track the absorption path of water. Figure 4b clearly shows a conspicuous color variance with the depth and time by detecting Raman band ratio of O-H/C-H bond, which decreases gradually in the depth direction at 200 min (Fig. 4c), explicitly verifying a distinct water gradient with depth. The top surface of PVA-AlgNa-based supramolecular hydrogel also demonstrates a more salient color change than that of PVA (Supplementary Fig. 17), showing superior moisture uptake capability. More importantly, the water gradient with depth still exists even after 1 week (Supplementary Fig. 18), which is a crucial prerequisite to form a sustained electric output. The long-standing water gradient mainly stems from strong water adsorption energy of PVA-AlgNa-based supramolecular hydrogel with asymmetrical absorption as indicated by DFT calculations.

On this basis, we further uncover the underpinning mechanism of moisture-initiated electric generation through deep investigation of dynamic water and ion diffusion process. 1D FTIR spectroscopy in

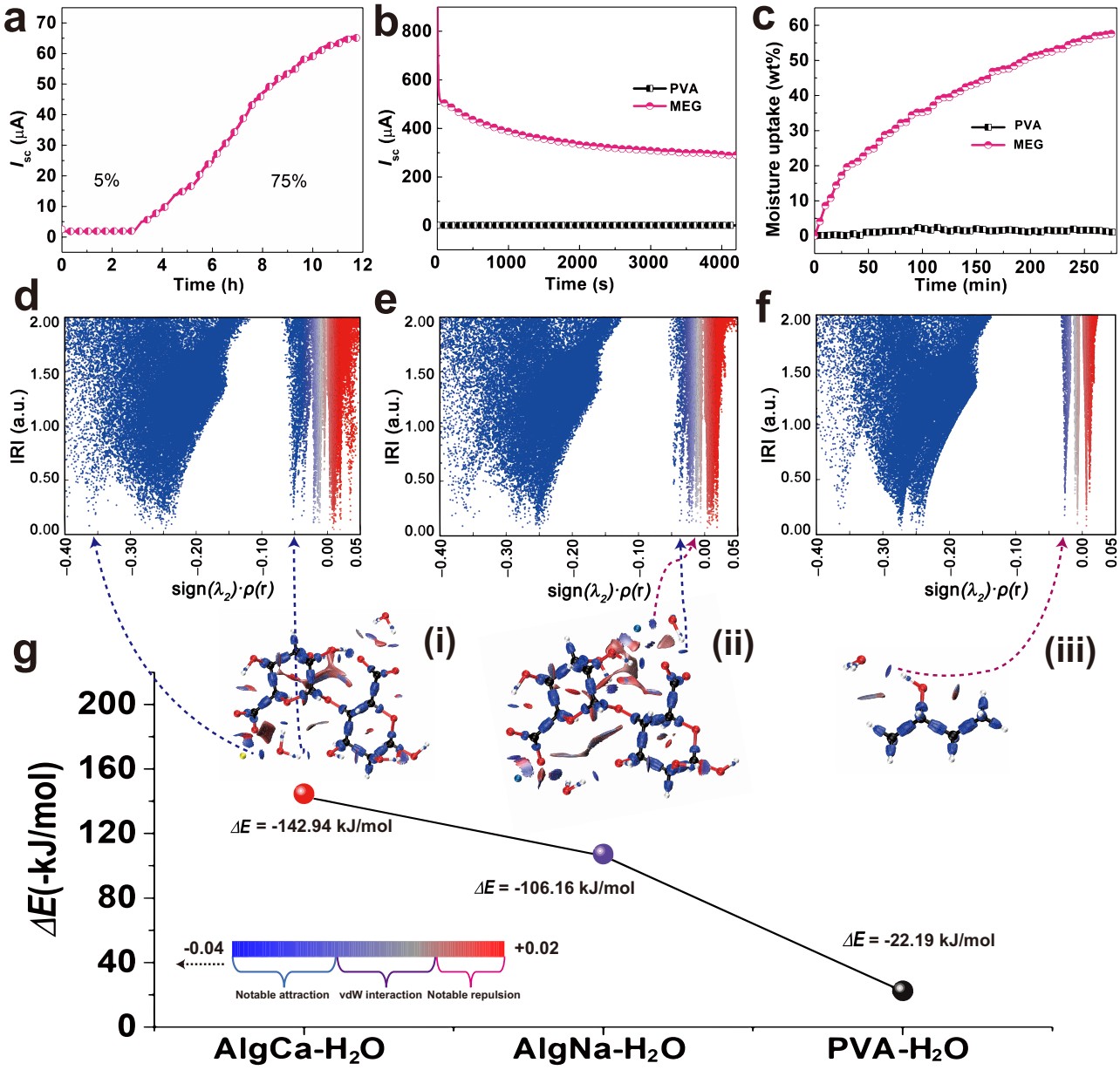

**Fig. 3 | The influencing factors and density functional theory (DFT) calculations. a** Evolution of $I_{sc}$ curve with RH change from 5% to 75% versus time. **b** $I_{sc}$ curves of MEG and PVA with the time (65% RH). **c** The moisture uptake capability with time of MEG and PVA (65% RH). Scatter maps between IRI (interaction region indicator) and sign($\lambda_2$)$\rho$ of AlgCa-$H_2O$ (**d**), AlgNa-$H_2O$ (**e**), PVA-$H_2O$ (**f**). **g** IRI maps and the corresponding interaction energies (kJ mol$^{-1}$) of AlgCa-$H_2O$ (i), AlgNa-$H_2O$ (ii), PVA (iii). The sign($\lambda_2$)$\rho$ is mapped on the isosurfaces. The arrows point to the corresponding sign($\lambda_2$)$\rho$ with different interaction strength between polymers and $H_2O$. The colors in the scatter maps (**d**–**f**) represent by the color bar in (**g**). Source data are provided with this paper.

Fig. 4d shows the strength of ν(–OH) gradually increases accompanied with the absorption of moisture. Based on this time-resolved 1D FTIR, 2D-FTIR spectroscopy is got by decoupling overlapped bands of O-H stretching band (ν(–OH)), which reveals water diffusion and water gradient construction from the molecular level. Figure 4e shows a positive auto peak in the synchronous spectrum, indicating that the ν(–OH) of water strengthens with time. Figure 4f displays the ν(–OH) in 1D FTIR spectra is deconvoluted into one positive cross-peak (3540, 3304 cm$^{-1}$) and one negative cross-peak (3304, 3115 cm$^{-1}$) in upper left triangle of asynchronous spectrum. According to the Noda rules[40,41], the specific sequence orders of bands mentioned above are 3540 > 3115 > 3304 cm$^{-1}$, which agrees well with the corresponding water diffusion coefficients calculated by Fickian diffusion equation[42] as shown in Supplementary Fig. 20. That means water diffusion follows

the sequence: weak bound water (3540 cm$^{-1}$), strong bound water (3115 cm$^{-1}$), cluster water (3304 cm$^{-1}$). Within MEG, weak bound water with small size and weaker hydrogen bonding takes the lead in diffusing to the bottom side. Then strong bound water moves more slowly because of its stronger hydrogen bonds and larger size[43]. Due to intense attraction force by AlgCa/Na network, water clusters are trapped around Na$^+$ or Ca$^{2+}$ ions to form ion water clusters, which will move until absorbing sufficient water to liberate ions. The smaller water diffusion coefficient of ion water clusters well verifies the relatively slow diffusion compared to that of other states of water. Therefore, MEG presents a unique behavior of "fast absorption" of moisture and "relatively slow diffusion" of ion water clusters[42,44]. By this way, distinct water gradient or ion concentration difference are spontaneously constructed between top absorbent surface and

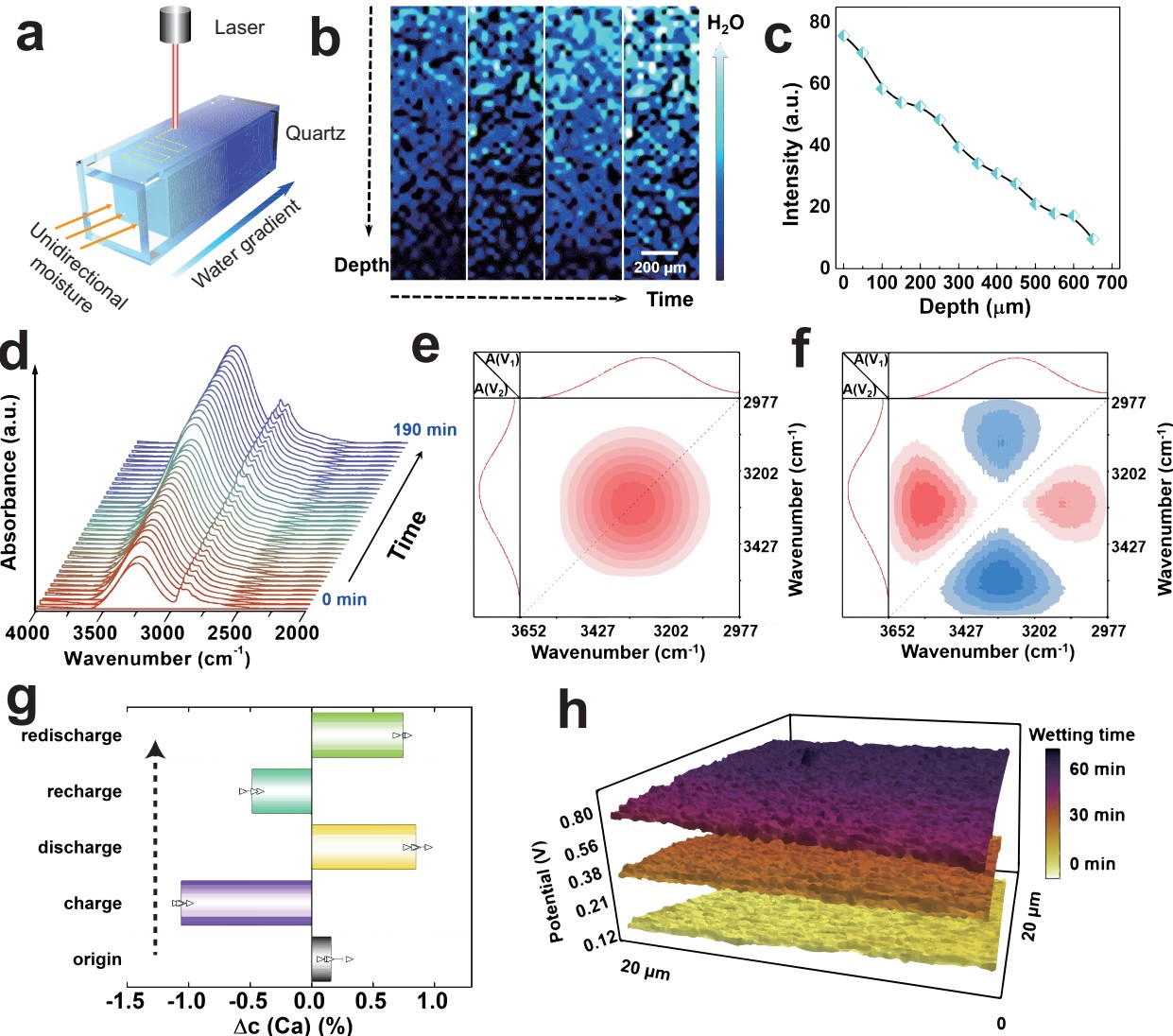

**Fig. 4 | Working mechanism of moisture-electric generator (MEG). a** Homemade set-up to measure the Raman spectrum of MEG sample. A transparent quartz tube is used to load MEG with only an open side to enable unidirectional moisture invasion. The yellow dotted line is the scanning route of laser. Water gradient is defined as the difference in moisture content between the upper side exposed to air and bottom side of the MEG sample. **b** The water diffusion with the depth and time revolution by 2D Raman mapping. The time for Raman spectroscopy mapping from left to right is 20, 70, 120, 200 min, respectively. The arrow for $H_2O$ represents more moisture is absorbed based on Raman band ratio O-H bond/C-H bond. **c** The normalized Raman band ratio of O-H/C-H bond changes with the depth at 200 min. The C-H bonds of stretching vibration area are within 2800–3000 $cm^{-1}$ and O-H bonds of stretching vibration area are within 3050–3650 $cm^{-1}$. **d** In situ FTIR (Fourier Transform Infrared Spectroscopy) spectrum tracking once the sample is exposed in the air (65% RH) versus time. 2D-FTIR correlation spectra in the 3700–2900 $cm^{-1}$ wavenumber region: **e** synchronous; **f** asynchronous contour maps. The red area and sky-blue area denote positive and negative correlation peaks, respectively. **g** Chemical component characterization of detective $Ca^{2+}$ variation between the top and bottom surface of MEG in different states by EDS. Data represent the mean ± standard deviation ($n = 4$). **h** The KPFM (Kelvin probe force microscope) detecting potential change of gel. The scan range is 20 × 20 μm. Source data are provided with this paper.

bottom barrier side for a long time, provoking a continuous and high electric output.

Furthermore, energy dispersive X-ray spectroscopy (EDS) is applied to examine ion diffusion under moisture. $\Delta c$(Ca) is defined as the difference of Ca element content between the top and bottom surfaces of MEG. A small positive value is observed in Fig. 4g at initial state. Then, the $\Delta c$(Ca) becomes negative after charging via capturing moisture from the air, implying that $Ca^{2+}$ ions transport from top to bottom surface. It is worth noting that the absolute value of $\Delta c$(Ca) becomes larger, which suggests that moisture enables extra dissociated $Ca^{2+}$ ions release from polymer chains (-COOCa) and then triggers these $Ca^{2+}$ ions migration. Whereafter, the MEG is discharged through short-circuit treatment. A reversed $\Delta c$(Ca) from negative into

positive is observed, indicating an opposite transport process. Apparently, the moisture-triggered ions migration directly contributes to the excellent electric generation of MEG, coupling with the joint contribution from directional migration of $Cl^-$, $Na^+$ (Supplementary Fig. 21). An analogous ion migration is obtained at the recharge-redischarge procedure. When extending the short-circuit time, $\Delta c$ values become smaller (Supplementary Fig. 22) and ion transport tends to establish an equilibrium. Meanwhile, the dissociated positive ions drift back to their initial place driven by the electrostatic attraction of bulky immovable negative $Alg^-$ chains. And the free positive ions tend to reunite the free $Cl^-$ ions. Nonetheless, such an ion-diffusion-induced current still outputs over hundred hours as shown in Supplementary Fig. 14b. In addition, Kelvin probe force microscope

(KPFM) is employed to probe the potential development with wetting (Supplementary Fig. 23). As displayed in Fig. 4h, the water-induced ion diffusion increases the potential from ca.150 to 720 mV. These results provide a straightforward and strong evidence that moisture-enabled ion movement is crucial to high power-output of MEG. On this basis, an underpinning mechanism is proposed (Supplementary Fig. 24). At the beginning of dried state, excess free ions gather at the top surface with restricted diffusion. Thus a relatively weak current and small voltage have been produced (Fig. 2d). As plenty of moisture are asymmetrically absorbed into upper surface of MEG, MEG obtains sufficient chemical conversion energy to dissociate ions from polymer chains. Meanwhile, an obvious water gradient is constructed with suspending ion water clusters. The sustained water gradient serves as driving force to provoke ion water clusters movement, then generating a high voltage of ca. 1.30 V. The hydration process reduces the internal resistance of MEG with triggered movable ions. With further short-circuit treatment, MEG generates a milliampere level current density on account of abundant ions back-diffusion. Combined with experimental results and DFT calculations, the synergistic effects of strong moisture-absorbing capability, long-standing water gradient, and then triggered abundant dissociated ion diffusion with enhanced ionic conductivity corporately empower the MEG excellent electric generation, compared with other sustained MEGs (Supplementary Table 4).

## Large-scale integration of MEGs

The scalability of MEG is considered as an effective method to promote MEG toward diversified application scenarios. The poor current output of one single MEG unit ($<10\,\mu A\,cm^{-2}$), reported previously, makes it difficult to achieve the practical application level ($>10$ mA) by several orders of magnitude despite in a large-scale integration[4]. Our newly developed MEG with satisfactory current output is highly promising to solve this thorny problem. As a proof of concept, an efficient and scalable integration process is developed (Supplementary Fig. 25). First, the bottom graphene electrodes were fabricated by directly laser patterning on flexible PI substrate efficiently and precisely. Then PEDOT:PSS layer was quickly stencil printed on the front end of bottom electrodes as carrier transport-assisting layer. Therewith, the gel was dripped on the bottom electrodes, followed by placing top electrodes and connecting next bottom electrode by head-to-end connection to achieve serial scalable integration. The parallel integrated ones are achieved by end-to-end and head-to-head parallel connection (Supplementary Fig. 26). The required current and voltage can be predetermined by the MEG bank design to satisfy the power requirement of practical applications.

Figure 5a shows that parallel-integrated MEG bank delivers steady $I_{sc}$ curves over 4000 s with a total voltage of ca. 1.3 V. $I_{sc}$ enlarges linearly with the parallel number of MEG units increasing from 10 to 280 units (Fig. 5b). What is thrilling is that the current is scaled up to ca. 65 mA with only 280 parallel units, which is one or two orders of magnitude better than previous integrated MEGs[43,45,46]. The average current density output is about 0.23 mA cm$^{-2}$ for each MEG unit (Supplementary Fig. 27). It is worth noting that only moisture stimuli under the room temperature is used to generate such high integrated current. To further demonstrate the stability of enlarged current, a 40-parallel-integrated MEG bank delivers a great current about 5.6 mA and the current outputs for over 24 h when connecting to external resistance of 33 Ω (Fig. 5c), which is further demonstrated by the photograph in Fig. 5f. The tens of milliampere electricity combined with upgraded voltage is sufficient to satisfy the power demanding of practical applications. As expected, the serial-integrated MEG bank also presents great linear scale-up performance (Fig. 5e). One single unit can output about 1.3 V and 50 serial units deliver a stable $V_{oc}$ of ca. 62.0 V for over 24 h (Fig. 5d, g). Furthermore, integrated MEG banks can directly charge commercial capacitors without the need of extra rectifiers. In Fig. 5h, capacitors with capacitance from 47 to 1000 μF

can be charged in less than 60 s by one single MEG unit. Capacitors of 47 μF are quickly charged up to 3.2, 4.9, and 15.3 V by serial MEG banks with 3, 4, and 12 units in Fig. 5i. Our MEG owns versatile, flexible, lightweight and scalable features, making it feasible as a DC power source to drive many IoTs devices in broad application scenes.

## Practical applications of MEGs

For real application scenarios, we pioneeringly employ integrated MEG bank to directly power many commercial electronics without auxiliary energy-storage devices and rectifying circuits. By virtue of flexible, lightweight, high-power-output and scalable merits, MEGs are successfully used to prepare self-powered electronics for real-life use. Figure 6a depicts that a MEG shirt worn on a sportsman can charge the smart watch at anytime and anywhere. Thus, the smart watch can get rid of dependence on cumbersome battery charger and keep working for a long time especially during outdoor activities. As a proof of concept in Fig. 6b and Supplementary Movie 1, an energy shirt integrated with 3*3 serial*parallel MEG bank is vividly demonstrated to charge a smart watch when needed. During charging, the temperatures of this MEG shirt remain almost the same as before charging (Fig. 6c), suggesting its facile working temperature when in close contact with human skins. More remarkably, a lamp bulb of 2.5 W is illuminated continuously by a large-scale MEGs bank with a 10*24 in parallel*serial combination as shown in Fig. 6d and Supplementary Movie 2. The predesigned MEG banks can render sufficient power output to drive optoelectronics for daily use. Beyond that, the MEG bank is small, lightweight, and flexible with sustained power output, promising the practical applications for IoTs. From the perspective of practical application, it is better to develop the power management system of the large-scale MEG bank for efficient energy harvesting and output in the future. As an example in Fig. 6e and Supplementary Movie 3, when driven by a small two-serial MEG bank, a LCD clock can run stably for over one month. Compared with the rigid and pollution-carrying battery, the green MEG bank provides an alternative power supply for small electronics. Furthermore, MEG banks can serve as gate voltage ($V_{gs}$) source to fabricate a self-powered metal-oxide-semiconductor field effect transistor (MOSFET) (Fig. 6f). Provided with a positive $V_{gs}$ of 6.0 V by integrated MEG banks, the drain current ($I_{ds}$) augments with external drain voltage ($V_{ds}$), indicating a switch-on state of MOSFET (Fig. 6g). While a switch-off status of MOSFET is shown under a negative $V_{gs}$ of −6.0 V. At a specific $V_{gs}$ in Fig. 4h, $I_{ds}$ of MOSFET displays a linear growth region firstly and then gradually climbs up to a saturated platform with increased $V_{ds}$. The similar phenomena happen on varied $V_{gs}$, suggesting the typical n-type performance of MOSFET. The above results manifest that the MEGs have been successfully employed to modulate MOSFET as a power source. These practical applications implicate great potential of our MEGs with versatile, high-power-density and easy-to-scale merits as a direct and sustainable energy source for broad IoTs applications.

## Discussion

In summary, a green MEG with high power output has been successfully developed by molecular engineering ecofriendly PVA-AlgNa-based supramolecular hydrogel as active material. One single unit with 1 cm$^2$ is capable of achieving an excellent power density (ca. 0.11 mW cm$^{-2}$) under only moisture stimuli and room temperature. Besides, a 9-cm$^2$-size MEG unit is able to generate both high $V_{oc}$ of ca. 1.30 V and sufficient $I_{sc}$ of ca. 2.14 mA, showing great potential for practical applications. Theoretical study and experimental results well verify that the enhanced moisture-capturing capability, remained water gradient, and then enabled dissociable ions migration bestowed by the supramolecular hydrogel empower MEG excellent electric generation. Notably, an unprecedented output current of ~65 mA is realized by a parallel MEG bank. As expected, scalable MEG devices as a DC power source greatly facilitate real-life applications, such as charging smart watch,

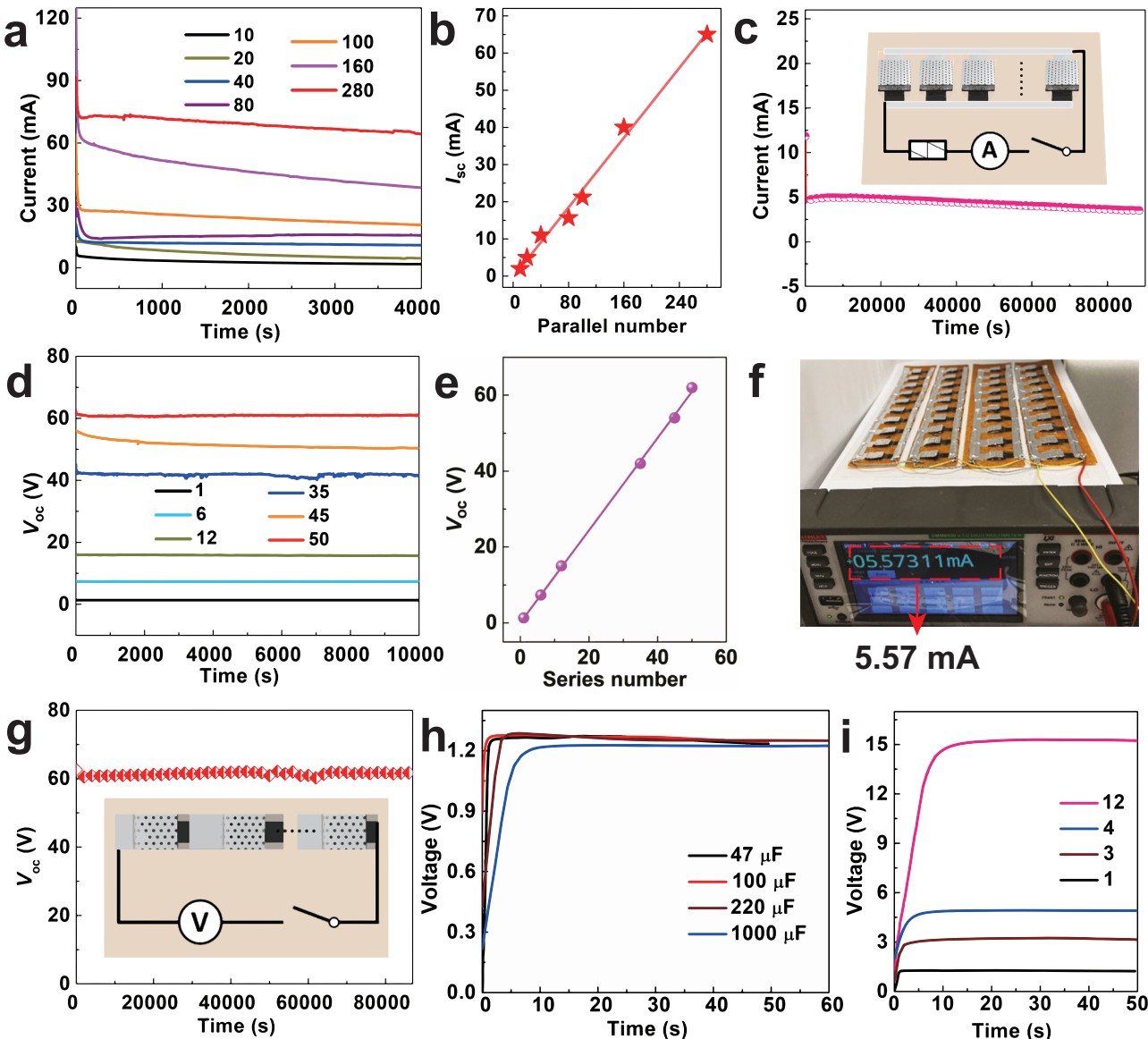

**Fig. 5 | The scalability of moisture-electric generators (MEGs) and charge-up for capacitors. a** The $I_{sc}$ curves of integrated MEGs with different parallel units at 80% RH. **b** The plot of $I_{sc}$ related to the parallel number of MEG units. The analytic linear fit equation is $y = -0.27 + 0.23*x$, R-square is 0.99. **c** The sustained current curve of 40-parallel-integrated MEGs with external load of 33 Ω for a long time. The inset shows the circuit diagram of parallel-integrated MEG bank connecting to external resistance. **d** The $V_{oc}$ curves of integrated MEGs with varying serial units. **e** The plot of $V_{oc}$ related to the serial number of device units. The analytic linear fit equation is $y = 0.04 + 1.22*x$, R-square is 0.99. **f** Photograph of 40-parallel-integrated MEGs connected to a load of 33 Ω, generating a sustained current of 5.57 mA at 65% RH. **g** The stable $V_{oc}$ curve with time for 50-serial-integrated MEGs for a long time. The inset shows the circuit diagram of serial-integrated MEG bank. **h** Voltage–time curves of different commercial capacitors (47, 100, 220, 1000 μF), directly charged by one MEG unit. **i** Voltage–time curves of a commercial capacitor (47 μF) charged by serial MEG banks with 1, 3, 4, and 12 units. Source data are provided with this paper.

illuminating a household bulb, driving a LCD clock continuously. This work charts the course towards the burgeoning development of green and high-performance MEG as sustainable and versatile energy source for broad IoTs applications and wearable applications.

## Methods

### Materials
PVA ($M_w$: 61,000) was purchased from Sigma Aldrich Co., Ltd. Glycerol (ACS, 99.5%) and calcium chloride ($CaCl_2$) anhydrous was offered by Shanghai Macklin Biochemical Co., Ltd. Sodium alginate (AlgNa) (200–500 mPa.s) was bought from Shenzhen Dieckmann Tech Co., Ltd. Poly(3,4 ethylenedioxythiophene) polystyrene sulfonate (PEDOT:PSS, 5.0 wt. %, conductive screen printable ink) was supplied by Sigma Aldrich Co., Ltd. The PI membrane was kindly provided by

Changchun Gao Qi polyimide material Co., Ltd, China. All reagents were used without further treatment.

### Preparation of MEG units
Firstly, laser printing of bottom electrode. As shown in Supplementary Fig. 25, the PI membrane firstly went through quick laser printing with optimized laser power and writing speed (12 W and 300 mm s⁻¹, respectively), which formed integrated graphene electrode as bottom electrode[47]. After that, PEDOT:PSS was stencil printed on bottom electrode as the mediation of electrodes/materials interfaces and carrier transport-assisting layer, which is dried at 50 °C in the oven. The thickness of graphene layer is about 10 μm and that of PEDOT:PSS is less than 1 μm. The resulting bottom electrode has repeatable and low electrical resistance of ca. 18.1 Ω/□.

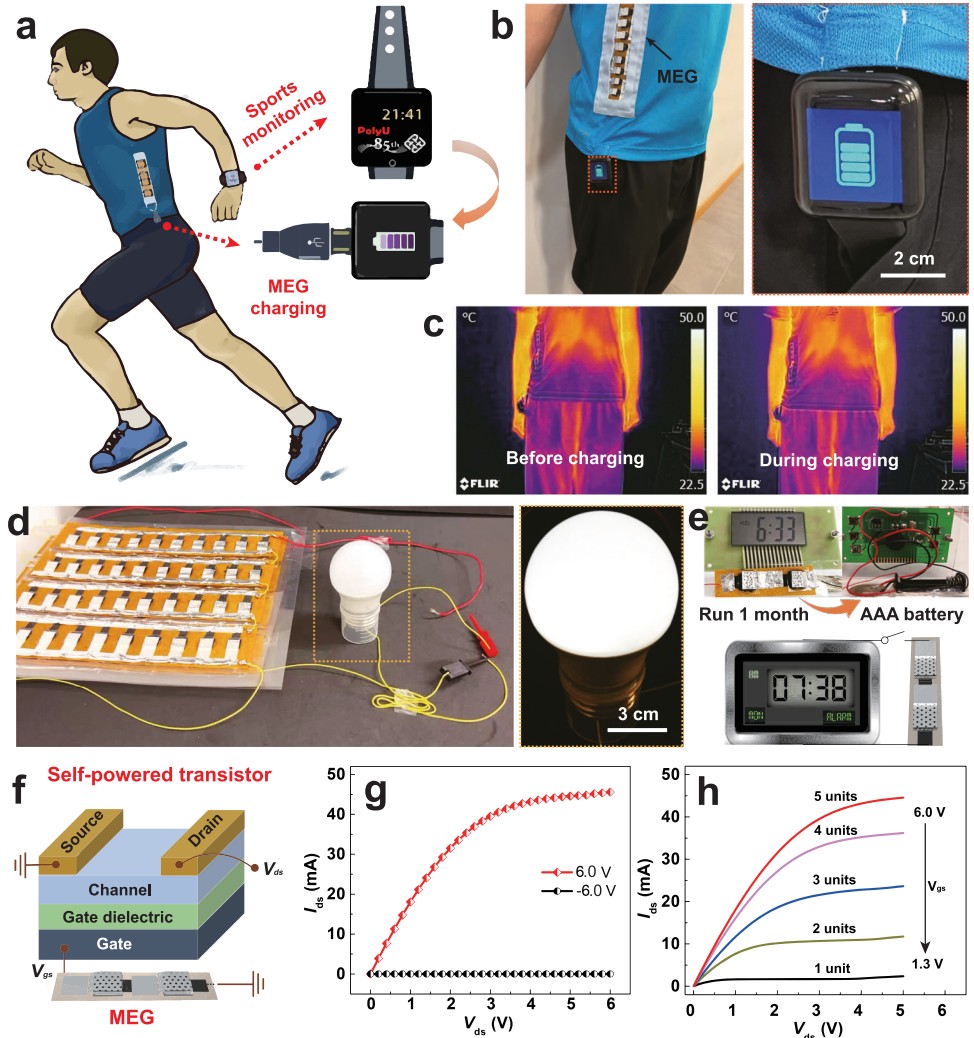

**Fig. 6 | Practical applications of moisture-electric generator (MEGs).**
**a** Illustration shows a sportsman wearing MEG sports shirt with a micro-USB to charge smart watch for sports and health monitoring. The orange arrow represents the smartwatch can be charged by MEG arrays after the smartwatch's battery runs out. MEG is defined as moisture electric generator. **b** Photograph of a shirt integrated with a 3*3 MEG bank (gray dotted box) charging a smart watch (orange dotted box, inset on right). **c** Temperature records of shirt integrated with MEG bank before charging (left) and during charging (right) smart watch. It shows that the facile working temperature of MEG bank when in close contact with human skins. **d** A lamp bulb (2.5 W) driven by integrated MEG bank. The inset with yellow dotted box on right shows lightened lamp bulb. **e** A LCD (liquid crystal display) clock powered by a 2 serial MEG bank for 1 month. **f** Diagram of the self-powered MOSFET (metal-oxide-semiconductor field effect transistor) driven by serial MEG devices. **g** Typical output characteristics of the MOSFET under the gate voltage of +6.0 V and −6.0 V supplied by 5-serial MEGs. **h** Typical output curves of self-powered MOSFET at varying gate voltage (1.3–6.0 V), driven by different serial MEG banks with 1, 2, 3, 4, 5 units. Source data are provided with this paper.

Secondly, preparing the active hydrogel layer. In all, 3 g PVA and 0.3 g AlgNa were added into 10 g DI water and dissolved at 95 °C for 2 h. Then 8 g glycerol was added into above mixture to keep stirring. At the same time, 0.7 g $CaCl_2$ was added into 7 g DI water stirring at room temperature until mixing uniformly. After that, the finely disseminated $CaCl_2$ was added into PVA/AlgNa mixture for further mixing until achieving a homogeneous solution. The homogeneous solution was drop-casted onto the well-prepared bottom electrode in advance with pre-designed thickness and size. A typical MEG device owns the area of 1 cm² and the thickness of 2 mm.

Thirdly, the connection of top electrode. An aluminum electrode was placed and adhered on the top of the hydrogel of the MEG, which went through fully drying and curing before testing. MEG with different AlgNa contents, $CaCl_2$ contents and different sizes were prepared by the same process. As the control sample, PVA was prepared by the same protocol. The films of PVA- and PVA-AlgNa-based hydrogel were drop-casted with 500 μm thickness for FTIR, 2D Raman top and bottom scanning, and KPFM tests.

## Characterization and measurement

**Electric measurement.** $V_{oc}$, $I_{sc}$, current and voltage with external loads, and capacitor charging curves were tested by Keithley 2400 (Tektronix, USA). Electrical characteristics of nMOSFET (SD-210) was measured by Keithley 4200 SCS. An environmental chamber was employed to regulate the temperature from −25 to 65 °C meantime with the RH kept at 80%. The different RHs were also regulated by the environmental chamber at the room temperature. All other RH tests were carried out at room temperature unless otherwise stated.

**Chemical component analysis.** The morphologies and structure of samples were characterized by SEM (VEGA3 TESCAN, Czech) and component analysis of samples was implemented by corresponding EDS. To test chemical component change of one MEG unit after charging or discharging process, the detailed procedure was described as follows. Firstly, one fresh-prepared MEG was dried for over 1 week to eliminate the influence of water. The top and bottom surfaces of gel within MEG were partially cut to test the original chemical component

by EDS. Secondly, the MEG was connected with the external circuit at open circuit state for ca. 5 h to get charged. The gel was further partially taken out to do the examination of both top and bottom surfaces by EDS. Thirdly, the MEG went through discharging process at short-circuit state for ca. 5 h, and the corresponding chemical component change was measured again using EDS. Whereafter, the rest of MEG underwent recharging process as the same as the first charging process and subsequent recharge–redischarge process. The corresponding chemical content variance was recorded by EDS in sequence. The test condition was at 65% RH and 22 °C.

**In situ monitoring moisture absorption and diffusion process.** 1D Raman spectrum was acquired by Renishaw Micro-Raman Spectroscopy System and a green LED laser with 532 nm used. To figure out the water gradient and ion diffusion under depth, 2D Raman mapping was implemented along with the depth of MEG sample. The hydrogel sample (depth about 2 cm) was prepared and fully dried at least 1 week in advance within a quartz tube. And the laser (532 nm) scans the dried sample from the top to bottom every 15 min once one side was exposed in the air under 65% RH. FTIR spectrometer (Nicolet iS50 FTIR Spectrometer, Thermo Scientific, USA) in attenuated total reflection (ATR) model was employed to measure chemical structures of samples with the scanning range of 650–4000 cm$^{-1}$ and the resolution of 4 cm$^{-1}$ over 32 scans. The water diffusion process was in situ tracking by time-resolved ATR-FTIR. Before testing, the samples were vacuumed and dried for 2 days first. Then the top surfaces of samples were exposed to the air (65% RH) and spectra were collected every 5 min. 2D correlation analysis was used for further in-depth analysis of water diffusion process at the molecular level. Before performing the 2D correlation analysis, the linear baseline corrections were processed in the regions of 3700–2900 cm$^{-1}$ firstly. Then 2D correlation ATR-FTIR spectra were plotted by 2D correlation spectroscopy software: 2DCS 4.0, developed by Zhou[41]. In 2D FTIR spectra, sky-blue and red cross peaks in the contour maps denote negative and positive correlation peaks, respectively. KPFM was conducted on a Scanning Probe Microscope (Asylum MFP-3D Infinity). Forward Looking Infra-Red camera (E33, FLIR, USA) was used to capture the temperature of MEG textile before charging and during charging.

## Theoretical study

DFT calculations in this study were performed with the Gaussian 09 d suite of programs. The structures of PVA, AlgNa, AlgCa, PVA-$H_2O$, AlgNa-$H_2O$, and AlgCa-$H_2O$ were optimized in gas phase at B3LYP/6-311G (d, p) level. The absorption energy ($\Delta E$) between $H_2O$ and polymers were calculated at the same level. The quantitative analysis of ESP on vdW molecular surface for the compounds was calculated by Multiwfn 3.8 program[48]. The color mapped isosurface graphs of ESP were rendered by VMD 1.9.3 program. The vdW surface referred throughout this paper suggests the isosurface of $r$ = 0.001e bohr$^{-3}$. The quantitative analysis of molecular surface is important to study non-covalent interaction. Firstly, electron density can be calculated by equation below[48]:

$$\rho(r) = \sum_i \eta_i |\varphi_i(r)|^2 = \sum_i \eta_i \left| \sum_I C_{I,i} \chi_I(r) \right|^2, \tag{1}$$

wherein $\varphi$ and $\eta$ represent the natural orbital and its occupation number, respectively. $\chi$ is basis function, and $C$ is coefficient matrix.

ESP can be represented as:

$$V_{\text{Total}}(r) = V_{\text{Nuc}}(r) + V_{\text{Elec}}(r) = \sum_A \frac{Z_A}{\sqrt[2]{(r - R_A)^2}} - \int \frac{\rho(r')}{\sqrt[2]{(r - r')^2}} dr' \tag{2}$$

where $Z$ and $\boldsymbol{R}$ mean nuclear charge and nuclear position, respectively. Then the $V_S^+$, $V_S^-$, and $V_S$, on behalf of average of positive, negative, and overall ESP on van der Waals (vdW) surface, respectively, can be calculated as expressions below:

$$V_S^+ = \left(\frac{1}{m}\right) \sum_{i=i}^m V(r_i) \tag{3}$$

$$V_S^- = \left(\frac{1}{n}\right) \sum_{j=1}^n V(r_j) \tag{4}$$

$$V_S = \left(\frac{1}{z}\right) \sum_{k=1}^n V(r_k) \tag{5}$$

where $i$, $j$, and $k$ are index of sampling points in positive, negative, and entire regions, respectively. $t$ is the total number of surface vertices. A positive (negative) value means that current position is dominated by nuclear (electronic) charges.

Interaction region indicator (IRI) analysis are able to exhibit various kinds of interaction regions in real space by combining with electron density ($\boldsymbol{\rho}$) and the sign of second eigenvalues of the electron-density Hessian matrix (sign($\lambda_2$)).

IRI is defined as follows:

$$\text{IRI}(r) = \frac{|\nabla \rho(r)|}{[\rho(r)]^{1.1}} \tag{6}$$

where $|\nabla \boldsymbol{\rho}|$ is the gradient norm of electron density:

$$|\nabla \rho(r)| = \sqrt{\left(\frac{\partial \rho(r)}{\partial x}\right)^2 + \left(\frac{\partial \rho(r)}{\partial y}\right)^2 + \left(\frac{\partial \rho(r)}{\partial z}\right)^2} \tag{7}$$

IRI is essentially the gradient norm of electron density weighted by scaled electron density. The isosurfaces of IRI are able to exhibit various kinds of interaction regions. The color mapped isosurface graphs of IRI were rendered by VMD 1.9.3 program[49].

## Ethical considerations

This study complies with all relevant ethical regulations and was approved by the Hong Kong Polytechnic University Institutional Review Board, protocol HSEARS20240116002. Researchers have obtained informed consent for publication of the images.

## Reporting summary

Further information on research design is available in the Nature Portfolio Reporting Summary linked to this article.

## Data availability

Relevant data supporting this study are available within the article and the Supplementary Information file. Source data are provided with this paper[50].

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

## Acknowledgements

This research was supported by the Research Grants Council of Hong Kong, China (No. 15201922E, 15203421E, 15202020E, 15201419E), Innovation and Technology Commission (MRP/020/21), Endowed Professorship Fund, The Hong Kong Polytechnic University (No.847A), and postgraduate scholarships by the Hong Kong Polytechnic University. The authors thank the technicians Y. K. CHAN, C. H. Tang, C. Y. MANG from Industrial Center of Hong Kong Polytechnic University for technical support. The authors also thank Prof. Zhou Tao from Sichuan University for 2D-FTIR analysis. The numerical calculations in this paper have been done on the supercomputing system in the Supercomputing Center of University of Science and Technology of China.

## Author contributions

X.T. and S.Y. conceived the concept. X.T. supervised the project. S.Y. designed and conducted the experiments. L.Z. contributed to DFT calculation and theoretical analysis. J.M. and J.H. contributed to measurement of Kelvin probe force microscope. J.G. and Y.C. contributed to transistor test. S.Y. wrote the original draft. X.T., W.C., and S.Y. reviewed and edited the manuscript.

## Competing interests

The authors declare no competing interests.
