## [Peer Review File · Nature Communications]

REVIEWER COMMENTS

Reviewer #1 (Remarks to the Author):

This manuscript reported moisture-electric generator (MEG) by using polyvinyl alcohol-sodium alginate hydrogel, enabling to generate an open-circuit voltage of 1.3 V and current density of 1.3 mA cm⁻². Based on fast absorption and slow diffusion, the hybrid hydrogel material simultaneously performs enhanced moisture absorption and remained water gradient, thus the MEG exerts high-performance and continuous output for long time. However, the electrode of MEG device is involved in active electrode material (Al). It is high concerning whether the remarkable electricity output of the MEG device is mainly attributed to an irreversible chemical reaction of Al component at the interface rather than moisture-induced ions diffusion. The authors need to provide the electricity output (voltage and current) of the MEG device by using inert electrode (Au, Pt, graphite). When the inert electrode substitute for active electrode, whether the electricity generation performance of MEG can still be unchanged? The authors need to be systemically demonstrated by experiments.

This work also lacks explanation for adopting asymmetric electrodes (Al-C). It is necessary to compare the electric output of the MEG based on symmetric electrodes (Al-Al, C-C) and asymmetric electrodes (Al-upper electrode-C bottom, C-upper-Al-bottom), and discuss the influence of electrode structure. In addition, this manuscript display that the device enables to continuously generate electricity for 17 hours (Figure 1e), but the authors propose that the MEG perform "stable electric output", which is a not rigorous elaboration. How the electricity-generation performance of the MEG device varies after 17 hours? Cyclic performance of MEG should be analyzed. Therefore, above key issues should be articulated for reviewer to warrant its consideration for significance of this work.

Other detailed and technical comments are added as follows:

1. In this work, the author mentioned "Slow diffusion" enables trapping water nearby to maintain water gradient through asymmetric absorption of moisture. However, the article does not quantitatively analyze the water molecules diffusion ability (e.g., diffusion coefficient) of hydrogel material. Besides, whether slow diffusion affects ion transport?
2. The power output of state-of-arts MEGs is mainly decided by water capturing ability and constructed water gradient. In terms of material design, how can the designed material balance high moisture absorption with a relatively sustainable moisture gradient? What characteristics do the materials require? Summarize a general strategy.
3. The polyvinyl alcohol-sodium alginate hydrogel is used as the material of electricity generation. As a type of hydrogel material, the hybrid hydrogel acquires a certain water content. How to control the initial water content of hybrid hydrogel? And what is the effect of the initial water content on the electric output?
4. Figure 3a-c show the moisture uptake capability and electric output of MEG. Furthermore, it is necessary to supplement the monitoring of the moisture uptake capability and electric output in real time during the electrical generation, which could be beneficial to propose the correlation between moisture absorption process and electricity generation.
5. Figure 2e exhibits the current output ($\sim 150 \mu\text{A}$) of MEG at 10% RH. What is the moisture uptake capability and water gradient of MEG at 10% RH? What is the current performance if Al electrode is replaced by an inert electrode? The authors did not explain why the MEG provide such a high current even at low humidity.
6. The authors need to systematically compare this hybrid hydrogel with other reported moisture-generating materials in terms of moisture absorption as well as ion dissociation and ionic conductivity.

Reviewer #2 (Remarks to the Author):

1. First and most important, though the power output performance reported in this paper is considerable, the potential influence of the Aluminum electrode corrosion is not clear. As the aluminum film was used as the top electrode, electrode corrosion due to the contact between water and metal is inevitable. It can also lead to a result of power generation. In fact, by using the

electrode corrosion caused the Aluminum electrode, even the generator using a layer of water-soaked tissue as the conductive medium can produce a voltage output of 1.2 V. The electricity generated by the oxidation of metals with air belongs to the field of metal-air batteries, such as the Al-air batteries in the Power Sources 437 (2019) 226896. Besides, compared to other electricity generation methods due to the electrode corrosion, the power generation performance in this paper has no advantage.

Therefore, to reveal the influence of supramolecular hydrogel, the inert electrodes or carbon materials should be used instead of Al electrode in the power generation system, and the results of power generation should be provided and compared with previous studies. If the authors cannot provide the experimental results with inert or carbon electrodes, this paper should be rejected.

2、As mentioned in the results and discussions, "As a natural polysaccharide, the polyanionic AlgNa features with numerous hydroxyl groups," "supramolecular AlgNa/Ca ionically crosslinked network is formed with abundant carboxyl functional groups (e.g., -COONa and -COOCa) in Fig. 1a." The FTIR spectra of AlgNA and polyvinyl alcohol (PVA)-sodium alginate (AlgNa) are suggested to be provided.

3、The moisture-absorption isotherm of the AlgNA and polyvinyl alcohol (PVA)-sodium alginate (AlgNa) are suggested to be provided, to evaluate the electrical energy output performance under different humidity conditions.

4、To better explore the influence of environmental humidity on the power generation performance, humidity response should be tested. The instantaneous change in ambient humidity should be introduced to observe charging of performance outputs.

5、As mentioned in the Working mechanism of MEG, "In addition, the current is immediately promoted after relieving the vacuum"" Fig. 3a shows an extremely low current for MEG at low RH of 5%, mainly due to few ion dissociation. The current gradually increases after exposing MEG at 75% RH with abundant ions diffusion." Is the main effect of the increase in ambient humidity to reduce the internal resistance of the device or to trigger plentiful ions diffusion? The impact of environmental humidity variations on voltage output should be discussed in detail, not only focusing on the changes in current.

6、The DFT calculation found that the interaction strength between H₂O and polymers decreases with the order: AlgCa > AlgNa > PVA. And, the power output of MEG is mainly decided by the constructed water gradient. Therefore, the experimental validation should be confirmed under the same ambient humidity and temperature that the sustained durability in electrical energy output as follows: AlgCa > AlgNa > PVA.

7、In Fig. 6, the application demonstration, more information about the energy management circuit should be given. How to maintain the stable voltage for the continuous work of the lamp bulb? How long does the generator array take (charging process) to accumulate the required the voltage level for the bulb?

Reviewer #3 (Remarks to the Author):

The manuscript reported a high-performance moisture-electric generator based on PVA-sodium alginate supramolecular hydrogel. The reported current and power outputs of the single unit MEG reach 1.31 mA cm⁻² and 110 μW cm⁻² with the open-circuit voltage of 1.3 V. These performance metrics exceed most of the existing literature by more than one order of magnitude. Such DC current keeps stable for more than 17 hours. The authors attributed the excellent performance of their MEG to enhanced moisture absorption and slow water diffusion. Towards practical applications, the authors demonstrated the highest short circuit of 65 mA through a parallel-connected MEG array with 240 units. A 2-serial MEG bank was able to power an LCD clock for one month.

Given the impressive short circuit current, long-term steady electrical output, and scalability of MEG, I think the PVA-alginate MEG will have a broad impact and attract readers from various fields. However, the authors did not provide enough convincing evidence to support the claim of slow water diffusion inside the PVA-alginate hydrogel. Therefore, I think significant revision is needed and the following concerns need to be resolved before this manuscript can be published in nature communication.

1. I think more experimental results or calculation results are needed to support the claim of slow water diffusion. The authors should clearly elaborate what is the water diffusion time in their MEG and how this value compares to previously reported humidity-driven electric generators.
2. I assume that the hydrogel conducts current through ion transport in the short-circuit condition. In this process, which ion dominates the process (e.g., Na⁺, Ca²⁺, Cl⁻ or the dissociated H⁺, OH⁻ ion from the water)? The ion transport is expected to establish an equilibrium in the short circuit condition. What is the time scale of such equilibrium in the MEG and will this ion equilibrium discourage the current output of MEG? The authors should provide some discussion on these points in their manuscript.
3. What is the error bar in Fig. 4g? I doubt the accuracy of using EDS to characterize the ion variation on the surface of the MEG. The observed variation is below 1%. However, to use EDS, the sample needs to be vacuumed in an SEM chamber, which I suppose will significantly influence the ion distribution on the sample surface. What is the area size of the EDS measurement to determine the Ca variation? Additionally, did the author measure the variation of other ions on the MEG surface using EDS?
4. In addition to my previous question, could the authors provide information on how the ion distribution on the surface of the MEG changes after it has been subjected to a steady short current condition for an extended period of time, for example, 1 week or 1 month?
5. What is the water concentration by weight percent when the MEG obtains the best energy conversion efficiency? How long does it need to obtain the optimum performance from a fully dried state? I assume that a fully dried MEG will have a very low short circuit current because of the dominance of ionic conductivity in the hydrogel. I think the internal resistance of the device in the dry and water-absorb states should also be provided.
6. What is the recharging time for the MEG unit? The authors have demonstrated a steady current output of 0.25 mA for 17 hours. Can current output at the level of 250 μ A be steady for a month without recharging?
7. The author analyzed the effect of CaCl₂ on the output current of the MEG device. Will the CaCl₂ influence the sustained output of the MEG device? I want to see this point.
8. The authors use the PVA with a molecular weight (MW) of \sim 65,000. Will the performance of MEG be influenced by the MW of the PVA? I want to see some performance comparisons of MEG with different PVA molecules (e.g., PVA with larger MW).
9. Could the authors provide some SEM images of their PVA-alginate hydrogel in both the fully-dried state and water-absorbed states?
10. For the 1.31 mA cm⁻² MEG, the surface area is only 0.01 cm². What is the thickness of this specific device? What is the accuracy of the surface area measurement? What is the long-term current performance of this device?
11. A plot of current density vs device area should also be provided in addition to Figure 2f.
12. Atoms in the ESP distribution of the PVA, AlgNa, AlgCa in Figure 1b should be labeled.
13. What is the freezing point of the PVA-alginate hydrogel? And what is the state of the PVA-alginate hydrogel at -25 °C? Pure PVA should freeze at -20 °C.
14. The color scale of Fig. S7 should be provided at least in A.U.
15. Figure 4d is not described and explained in the main text.
16. The current per unit area for the large-scale MEG array should also be provided.
17. Can the authors elaborate on the process by which the PVA-alginate solution underwent gelation? Was the freeze-thaw method used or did the solution gelate at ambient temperature?
18. Can the authors confirm if the asymmetric moisture absorption of the MEG is solely due to the asymmetric electrode design, or are there additional strategies implemented to promote this asymmetric moisture absorption?

We are very grateful for the time and effort made by the Reviewers, especially for their helpful suggestions and constructive comments. We have revised the manuscript and supporting information accordingly. The changes are highlighted in color. The detailed responses are given as follows:

RESPONSE TO REVIEWERS' COMMENTS

Reviewer #1 (Remarks to the Author):

This manuscript reported moisture-electric generator (MEG) by using polyvinyl alcohol-sodium alginate hydrogel, enabling to generate an open-circuit voltage of 1.3 V and current density of 1.3 mA cm⁻². Based on fast absorption and slow diffusion, the hybrid hydrogel material simultaneously performs enhanced moisture absorption and remained water gradient, thus the MEG exerts high-performance and continuous output for long time. However, the electrode of MEG device is involved in active electrode material (Al). It is high concerning whether the remarkable electricity output of the MEG device is mainly attributed to an irreversible chemical reaction of Al component at the interface rather than moisture-induced ions diffusion. The authors need to provide the electricity output (voltage and current) of the MEG device by using inert electrode (Au, Pt, graphite). When the inert electrode substitute for active electrode, whether the electricity generation performance of MEG can still be unchanged? The authors need to be systemically demonstrated by experiments.

[Author's reply]: Thanks a lot for your pertinent comments. To elaborate the effect of electrodes, different inert electrodes (Au, Pt, and C electrodes) have been used to prepare MEGs. And their electricity generation performance was tested to compare with that of MEG using Al electrode as shown in Fig. R1-1.

The results show that all MEGs with different inert electrodes yield favorable electric outputs, which are comparable with that of MEG using Al electrode. Fig. R1-1a demonstrates that MEGs using other inert electrodes output similar V_{oc} with MEG using Al electrode. For MEG using C electrode, a continuous I_{sc} output of about 350 μ A is maintained beyond 4000s at 80% RH as shown in Fig. R1-1b, approaching that of MEG using Al electrode. Fig. R1-1c and d show that MEGs using Pt and Au electrode can also deliver about 100 and 80 μ A over 4000s,

respectively. Such small difference for MEGs with different electrodes is acceptable and also observed in other references. [Nature Nanotechnology. 2021; 16: 811] [Energy & Environmental Science. 2023; 16: 2338] [Energy & Environmental Science. 2019; 12: 972] [Advanced Functional Materials. 2023; 33: 2211013] Based on the comparison of MEGs with different electrodes, it can be safely concluded that the supramolecular hydrogel is the key for electric generation of our MEG instead of electrodes. The supramolecular hydrogel enhances the moisture absorption of the MEG to promote the sufficient chemical conversion energy, consequently inducing high power output. Similar with other previous works, Al electrode is actually a common selection for MEGs. [Energy & Environmental Science. 2016; 9: 912] [Energy & Environmental Science. 2023; 16: 2338] Compared to the inert electrodes like Au, Al is highly flexible, lightweight, easily accessible and fairly cheap, which is desirable for scalable and low-cost MEGs towards wide applications.

Fig. R1-1 The electric generation performance of the MEGs assembled by different inert electrodes. (a) V_{oc} output of the MEGs with different electrodes at the same condition. The I_{sc} output of the MEGs with top electrode replacing by C electrode (b), Pt electrode (c), Au electrode (d) under 80% RH.

Furthermore, MEGs with C-C electrode were prepared to systematically measure the humidity

and temperature-dependent characteristics. Fig. R1-2a shows that the average I_{sc} augments monotonically from ca. 50 μA up to ca. 330 μA with the rising of RH from 10 to 80%. The accelerated current mainly derives from enhanced moisture absorption to trigger substantial ion diffusion. Besides, V_{oc} presents a gradual rise to ~ 1.2 V with the RH. In addition, our MEG also shows great adaptability under wide range of temperature. From 0 to 50 $^{\circ}\text{C}$, the average J_{sc} shows a sharp climb from ca. 15 to 552 μA (Fig. R1-2b). Such obvious growth benefits from the synergistic effect of accelerated ion transport rate and ion concentration at high temperature. The elevated temperature also enhances V_{oc} to about 1.2 V. The humidity and temperature-dependent characteristics of MEGs with C-C electrode display the same tendency with that of C-Al electrode based MEG. Overall, MEGs with inert electrodes demonstrate comparable electric output to MEGs with C-Al electrode, even changing humidity/temperature conditions. It further verifies moisture-induced ions diffusion is the driving force for electric generation of our MEG device.

Fig. R1-2 The I_{sc} and V_{oc} of MEG with C-C electrodes at (a) different RHs tested at room temperature and (b) different temperature tested at 80% RH.

The electric generation performance of MEGs assembled by different inert electrodes at different RHs and temperature has been added in the supporting information of Fig. S7 on page 9. The discussion is supplemented in the manuscript on page 9 as follows: “Furthermore, MEGs with different top electrodes output favorable electric performance and display similar humidity/temperature-dependent characteristics in Fig. S7, suggesting electricity generation mainly derives from moisture absorption by supramolecular hydrogel.”

This work also lacks explanation for adopting asymmetric electrodes (Al-C). It is necessary to

compare the electric output of the MEG based on symmetric electrodes (Al-Al, C-C) and asymmetric electrodes (Al-upper electrode-C bottom, C-upper-Al-bottom), and discuss the influence of electrode structure. In addition, this manuscript display that the device enables to continuously generate electricity for 17 hours (Fig. 1e), but the authors propose that the MEG perform “stable electric output”, which is a not rigorous elaboration. How the electricity-generation performance of the MEG device varies after 17 hours? Cyclic performance of MEG should be analyzed. Therefore, above key issues should be articulated for reviewer to warrant its consideration for significance of this work.

[Author’s reply]: Thanks a lot for your comments. The electric output of the MEGs based on symmetric electrode and asymmetric electrode structures are compared as shown in Fig. R1-3. The MEGs with symmetric/asymmetric electrode structures show similar V_{oc} outputs. MEG with top electrode (Al)-bottom electrode (C) structure shows slightly higher I_{sc} than MEGs with C-Al and C-C structures. The similar V_{oc} and comparable I_{sc} outputs indicate that the change of relative electrode positions exerts limited influence on electricity generation of the MEG.

Fig. R1-3 The electric generation performance of MEGs with different electrode structures (top electrode-bottom electrode: Al-C, C-Al and C-C) were tested at 80% RH.

It is believed that a pair of asymmetric electrode design is conducive to large-scale integration and improve output performance of MEG device. [Energy & Environmental Science. 2019; 12: 1848] Inspired by this, we give priority to adopting asymmetric electrodes instead of symmetric

ones. In fact, asymmetric electrode design, with C as bottom electrode and multi-hole Al film as top electrode, are chosen based on the following reasons. Firstly, the flexible bottom C electrodes can be fabricated by directly laser patterning on soft PI substrate efficiently and precisely on a large scale. [ACS nano. 2020; 14: 3219][Nature communications. 2014; 5: 5714] It is easy to realize the localization and scale-up for large integration of MEG arrays. Secondly, flexible and lightweight Al films can easily connect to adjacent C electrodes. Besides, Al film is highly conductive, easily accessible and fairly cheap, which is desirable for scalable and low-cost MEGs towards wide applications. The conductivity of Al film is not influenced by punching holes. In contrast, if asymmetric electrode with top electrode (C)-bottom electrode (Al) structure is adopted, it needs extra substrate to load Al films as bottom electrodes. Moreover, the conductivity of C electrodes will be impaired by punching holes process, which may result in slightly decreased current output as shown in Fig. R1-3. Thus, we give preference to asymmetric electrodes with Al-C structure.

We have added the explanation about the asymmetric electrode with top electrode (Al)-bottom electrode (C) structure in the manuscript on page 5 as follows: “Besides, a pair of asymmetric electrodes were established by directly laser-induced graphene as bottom electrode on polyimide (PI) substrate and aluminium (Al) film with holes as top electrode, which are conducive to large-scale integration efficiently and improve output performance of MEG device. Additionally, Al film is highly flexible, lightweight, easily accessible and fairly cheap, which is desirable for scalable and low-cost MEGs towards wide applications.^{27, 28}” The effect of asymmetric electrode structures on MEGs has been added in the supporting information of Fig. S9 on page 11. And the discussion is also added in the manuscript on page 9 as follows: “Besides, the change of relative electrode positions exerts limited influence on electricity performance of MEG (Fig. S9).”

Fig. 1e in the manuscript has shown that a large current output of ca. 0.25 mA keeps steady for about 17 hours when connected to an external resistance of 1 k Ω . Compared to many works with stable output in tens of mins, [Journal of Materials Chemistry A. 2021; 9: 7085] [Materials Horizons. 2021; 8: 2303] [Advanced Materials. 2022; 34: 2106410] [Nano Energy. 2022; 94: 106942] Our MEG can deliver a stable current of ca. 0.25 mA for about 1000 min. That proves

a relatively long stability of our MEG. To figure out the current output of MEG after 17 hours, a continuous current measurement of MEG was carried out over one month as shown in Fig. R1-4. After a steady current output of ca. 0.25 mA over 17 hours, a gradually deteriorated current in Fig. R1-4 is observed with the pass of time, which may derive from the saturation of water adsorption. Despite that, the inserts in Fig. R1-4 show the current can still output 8 μ A over one month, suggesting the excellent sustainability of our MEG accompanied by dynamic water adsorption-desorption exchange at the interface.

Fig. R1-4 The current curve of MEG over one month with the loading resistor of 1 k Ω under 70% RH and room temperature. The inserts show the current curves of the MEG device at different time slots.

Since water is the main energy source of MEG, it is conceived that the electricity regeneration can be realized by re-absorption from air after dehydration of MEG. Furthermore, the cyclic electric performance of the MEG was tested through water adsorption-dehydration-adsorption cycles. The resulting current curves are demonstrated in Fig. R1-5. The first absorption cycle enables MEG deliver a sustained current of about 0.22 mA. After that, MEG undergoes dehydration process by drying at 20% RH and room temperature about 24 h. The subsequent absorption cycles also drive MEG to generate sustained current output despite the current outputs become slightly smaller, displaying decent cycle electric performance of the MEG.

Fig. R1-5 Cyclic electric performance of MEG with load resistance of 1 k Ω in water adsorption-dehydration-adsorption cycles under 70% RH and room temperature.

For a more rigorous elaboration, “stable electric output” is modified into “sustained/continuous electric output” in the manuscript on page 4, 12, 15, 16, 17. Besides, long-time performance has been added in the supporting information of Fig. S3 on page 6 and discussed in the manuscript on page 7 as follows: “More significantly, a large current output of ca. 0.25 mA keeps steady for about 17 hours when connected to an external resistance of 1 k Ω (Fig. 1e). Beyond that, the current output gradually decays due to saturation of moisture absorption but persists for over one month (Fig. S3).” Furthermore, the cyclic electric performance has been added in the supporting information on page 13 in Fig. S12 and discussed in the manuscript on page 10 as follows: “In addition, a decent cyclic current output was also observed by moisture adsorption-dehydration process (Fig. S12). These results further indicate that moisture is the prime energy source of MEG.”

Other detailed and technical comments are added as follows:

1. In this work, the author mentioned “Slow diffusion” enables trapping water nearby to maintain water gradient through asymmetric absorption of moisture. However, the article does not quantitatively analyze the water molecules diffusion ability (e.g., diffusion coefficient) of hydrogel material. Besides, whether slow diffusion affects ion transport?

[Author’s reply]: Thanks a lot for your pertinent comments. In our work, “slow diffusion”

means that ion water clusters transport slower than other states of water (like weak bound water, strong bound water), which is decoupled from 2D-FTIR spectroscopy (Fig. 4e and f). By decoupling overlapped bands of O-H stretching band, the specific sequence orders of bands are $3540 > 3115 > 3304 \text{ cm}^{-1}$. That means water diffusion follows the sequence: weak bound water (3540 cm^{-1}), strong bound water (3115 cm^{-1}), cluster water (3304 cm^{-1}).

To quantitatively analyze the water molecules diffusion ability of hydrogel, we further extract quantitative information from the 1D-FTIR profiles by integrating the intensities of the 2D-FTIR patterns. Hence, the intensities of three decoupled peaks ($3540, 3115, 3304 \text{ cm}^{-1}$) with the time were obtained. Furthermore, equation 1 is given to estimate the effective diffusion coefficient of water from FTIR spectra based on the Fickian diffusion. [The Journal of Physical Chemistry B. 2008; 112: 2880]

$$\frac{A_t}{A_\infty} = 1 - \frac{8\gamma}{\pi[1 - \exp(-2\gamma L)]} \times \sum_{n=0}^{\infty} \left\{ \frac{\exp(g) [f \exp(-2\gamma L) + (-1)^n (2\gamma)]}{(2n + 1)(4\gamma^2 + f^2)} \right\} \quad (1)$$

Where

$$g = \frac{-D(2n + 1)^2 \pi^2 t}{4L^2}, f = \frac{(2n + 1)\pi}{2L}$$

In equation 1, A_t is the band absorbance of the FTIR spectra at time t , A_∞ is the band absorbance at equilibrium, γ is the penetration depth of the evanescent wave, L is the thickness of the polymer membrane (invariable), and D is the diffusion coefficient. Thus the diffusion coefficients of different states of water can be calculated by a nonlinear curve fitting [Macromolecules. 2002; 35: 5500] to equation 1 from the variation of the three decoupled peaks ($3540, 3115, 3304 \text{ cm}^{-1}$) versus time.

The calculated results are shown in Fig. R1-6. The D value for three peaks also follows the order: $3540 > 3115 > 3304 \text{ cm}^{-1}$, which agrees well with the results of 2D-FTIR spectroscopy. That means weak bound water diffuses in the fastest way, while ion water cluster diffuses in the slowest way due to intense attraction force by AlgCa/Na network. Thus, the “slow diffusion” here means “relatively slow diffusion” of ion water cluster compared to other states of water.

The ion transports as the form of ion water cluster. Thus ion transports also slow down with a smaller D value. Despite the relatively slow transport of ion water cluster, it is actually beneficial to maintain ion concentration gradient. On the other hand, it exerts little effect on the abundant ions dissociated from hydrogel due to plenty of moisture absorption. Therefore, MEG still delivers great electric output by virtue of strong moisture-absorbing capability, long-standing water gradient, and then triggered abundant dissociable ions.

Fig. R1-6 The integrated intensities of the three decoupled peaks versus time, (a) 3540 cm^{-1} ; (b) 3115 cm^{-1} ; (c) 3304 cm^{-1} .

We have modified the manuscript on page 3, 12, 14 about “slow diffusion” as “relatively slow diffusion”. Besides, the quantified water diffusion coefficients have been added in the supporting information on page 18 in Fig. S20. The discussion is added on page 15 in the manuscript as follows: “According to the Noda rules,^{40, 41} the specific sequence orders of bands mentioned above are $3540 > 3115 > 3304\text{ cm}^{-1}$, which agrees well with the corresponding water diffusion coefficients calculated by Fickian diffusion equation⁴² as shown in Fig. S20.....The smaller water diffusion coefficient of ion water clusters well verifies the relatively slow diffusion compared to that of other states of water. Therefore, MEG presents a unique behavior of “fast absorption” of moisture and “relatively slow diffusion” of ion water clusters.”

2. The power output of state-of-arts MEGs is mainly decided by water capturing ability and constructed water gradient. In terms of material design, how can the designed material balance high moisture absorption with a relatively sustainable moisture gradient? What characteristics do the materials require? Summarize a general strategy.

[Author's reply]: Thanks a lot for your useful suggestions. As we know, the power output of state-of-arts MEGs is mainly decided by water capturing ability and constructed water gradient. However, excessive captured water tends to reach equilibrium quickly. So it is challenging to achieve great water capturing and long-term maintained water gradient at the same time. Great efforts have been made to enhance moisture absorption capability by the physical-chemical modifications and creating hierarchical pores. Physical or chemical modifications like laser irradiation, plasma treatment or over-oxidization treatment are effective ways to improve the surface hydrophilicity of active materials. [Advanced Materials. 2017; 29: 1604972] [Nature Communications. 2018; 9: 4166] [Energy & Environmental Science. 2023; 16: 2338] The hierarchically pores can increase the specific contact area between active materials and water molecules. [Advanced Materials. 2015; 27: 4351][Advanced Energy Materials. 2022; 12: 2202634]

Besides, durable water gradient relies on not only built-in chemical gradient but also providing asymmetric moisturization. Chemical gradient can be constructed with hydrophilicity gradient or ion density gradient by thermal reduction, laser irradiation, polarization, or heterogeneous composites with different types of functional groups. [Nature Communications. 2022; 13: 3643] [Energy & Environmental Science. 2018; 11: 2839] [Nature Nanotechnology. 2021; 16: 811] Asymmetric moisturization can be realized by unidirectional moisture stimuli or asymmetric relative humidity environment. [ACS Materials Letters. 2021; 3: 193]

To achieve high moisture absorption with a relatively sustained moisture gradient, it is better to combine above strategies. For example, heterogeneous composites with different wettability can promote self-sustained electric generation by the efficient absorption, transmission, and evaporation of water. [Nature Communications. 2022; 13: 3643] Also, materials with high porosity and ion density gradient not only improve the contact area but also construct remained water gradient, leading to an enhanced electricity. [Advanced Functional Materials. 2022; 33: 2210027]

Moreover, we have added the discussion about the general strategy in the introduction of revised manuscript on page 3 as follows: "Great efforts have been made to enhance moisture absorption capability by the physical-chemical modifications to improve surface

hydrophilicity^{7, 18} and creating hierarchical pores to increase specific contact area between active materials and water molecules.^{3, 19} Additionally, durable water gradient relies on not only built-in chemical gradient by thermal reduction, laser irradiation, polarization, or heterogeneous composites,^{9, 20, 21} but also providing asymmetric moisturization by unidirectional moisture stimuli or asymmetric humidity environment.⁴ The feasible strategies are to combine above methods to achieve high moisture absorption with a sustained moisture gradient. For example, heterogeneous composites with different wettability can promote self-sustained electric generation by the efficient absorption, transmission, and evaporation of water.²⁰ Also, materials with high porosity and ion density gradient not only improve the contact area but also construct remained water gradient, leading to an enhanced electricity.^{22,}

3. The polyvinyl alcohol-sodium alginate hydrogel is used as the material of electricity generation. As a type of hydrogel material, the hybrid hydrogel acquires a certain water content. How to control the initial water content of hybrid hydrogel? And what is the effect of the initial water content on the electric output?

[Author's reply]: Thanks a lot for your kind comments. The water content of hybrid hydrogel can be controlled by drying conditions, including changing RH and temperature. After 24 hours' gelation, the hybrid hydrogels went through another 12 hours' drying. For the first drying method, the samples were dried at different RHs (65% and 20%) at room temperature. As shown in Figure R1-7a, the water content is about 28 wt% when samples are dried at 65% RH. In comparison, the calculated water content is about 24 wt% when the drying condition is 20% RH at room temperature, which is slightly smaller than that dried at 65% RH. Secondly, different temperatures (25, 50, 70 °C) with a fixed 20% RH have been employed to dry samples. Figure R1-7b displays that a gradual decrease of water content is observed from 24 wt% to 13 wt% with temperature rising from 25 to 70 °C.

Fig. R1-7 (a) The water contents of MEGs dried at 65% and 20% RH, respectively. The temperature is fixed at 25 °C. (b) The water contents of MEGs dried at 25 °C, 50 °C, 70 °C, respectively. The RH is fixed at 20%.

Furthermore, the electric outputs were tested for above samples once exposed at 80% RH and room temperature as shown in Fig. R1-8. Fig. R1-8a shows that MEG dried at 65% delivers a high V_{oc} of ca. 1.3 V and a large I_{sc} of ca. 0.4 mA, which keeps almost the same for MEG dried at 20% RH. Besides, the V_{oc} and I_{sc} of MEGs dried at 50 °C are competitive with those of MEGs dried at 25 °C, while is obviously larger than those of MEGs dried at 70 °C. The high temperature of 70 °C may cause contact issue between hydrogel and electrode, leading to a lower electric output. Based on above results, it is reasonable to deduce that the drying conditions at a wide range of 20-65% RH and 25-50 °C exert little impact on the final electric output of MEG despite the initial water content varies from 28 wt% to 16 wt%. For the underlying reason, strong moisture absorption capability of MEG guarantees that subsequent water capturing is not impacted at an unsaturated state. Thus the final electric output is also not affected when dried under suitable conditions.

Fig. R1-8 (a) The V_{oc} and I_{sc} for 20% and 65% dried MEGs, respectively. (b) The V_{oc} and I_{sc} of

MEG dried at 25, 50, 70 °C, respectively. The test condition is 80% RH and room temperature.

In addition, the drying condition as well as the electric output results are added in the supporting information of Fig. S6 on page 8. Meanwhile the manuscript is modified on page 9 as: “MEGs dried at suitable conditions show similar electric outputs after exposed in the same test environment (Fig. S6), indicating that the strong moisture absorption capability is critical to electric generation.”

4. Fig. 3a-c show the moisture uptake capability and electric output of MEG. Furthermore, it is necessary to supplement the monitoring of the moisture uptake capability and electric output in real time during the electrical generation, which could be beneficial to propose the correlation between moisture absorption process and electricity generation.

[Author’s reply]: Thanks a lot for your useful comments. To gain deep insight into the correlation between moisture absorption process and electricity generation, we have supplemented the in-situ current output measurement during weighting process. Fig. R1-9a clearly demonstrates that moisture absorption of MEG increases quickly at first and then slowly reaches a platform about 68 wt%, followed by a small fluctuation for a long term. It means that the moisture absorption of MEG reaches a saturation state with moisture adsorption and desorption dynamically. At the same time, a continuous short-circuit current output generates over 120 hours (Fig. R1-9b), which associates well with the moisture adsorption process. The continuous adsorption of water molecules couples with the ion dissociation process in the MEG, leading to a continuous current output even after reaching absorption saturation state despite a gradually decayed current. The possible reason lies in that the ion concentration is expected to establish an equilibrium over a long period. [Energy & Environmental Science. 2019; 12: 1848] It well demonstrates that moisture absorption process is directly related to electricity generation of MEG.

Fig. R1-9 Moisture uptake capability and current output of MEG versus time synchronously. (a) The mass increase of MEG by moisture adsorption versus time under 80% RH and room temperature. (b) The concurrent current output of MEG versus time at the same condition. (c-d) The tested current output of MEG at different time slot.

In addition, we have supplemented the moisture uptake capability and current output of MEG versus time synchronously in the supporting information of Fig. S14 on page 14. Besides, the discussion is added in the revised manuscript on page 11: “A sustained current lasts for more than 120 h accompanied by the concurrent adsorption of water molecules, which couples with the ion dissociation process in the MEG even after reaching absorption saturation state (Fig. S14). It well demonstrates that moisture absorption process is directly related to electricity generation of MEG.”

5. Fig. 2e exhibits the current output ($\sim 150 \mu\text{A}$) of MEG at 10% RH. What is the moisture uptake capability and water gradient of MEG at 10% RH? What is the current performance if Al electrode is replaced by an inert electrode? The authors did not explain why the MEG provide such a high current even at low humidity.

[Author’s reply]: Thanks a lot for your kind comments. To quantitatively analyze the moisture uptake capability at 10% RH, moisture-absorption isotherm of MEG has been tested by a

dynamic vapor sorption analyzer (BSD-DVS) at a constant temperature of 25 °C. Fig. R1-10a displays a gradual weight increment to a platform of ca. 2.1 wt% for MEG after exposing in the air over 180 min. The moisture uptake capability of MEG is about 2.1 wt% at 10% RH. It suggests that MEG is able to absorb moisture from the air even at a very low 10% RH. Furthermore, the moisture uptake capabilities at different RHs are shown in Fig. R1-10b. With the increase of RH, the moisture uptake capability gradually upgrades from 2.1 wt% to 90.0 wt% at 90% RH, which suggests superior moisture uptake capacity of MEG no matter of a low RH or a high RH. It is the strong moisture uptake capability that makes our MEG can absorb moisture at an extreme environment to trigger ion diffusion, thus generating a large current output.

Fig. R1-10 (a) The specific absorption kinetics curve under 10% RH of PVA-AlgNa hybrid hydrogel. (b) Moisture-absorption isotherm at different RHs for PVA-AlgNa based hydrogel. The test temperature is at 25 °C.

The water gradient can be constructed since MEG is able to absorb moisture from the air under 10% RH. Through Raman mapping (Fig. 4a-c in the manuscript), it is feasible to obtain semi-quantitative analysis of water gradient for MEG. While the sample chamber for Raman measurement cannot regulate the relative humidity at current experiment condition. We realize the difficulty of directly detecting water gradient at 10% RH. In the next work, we plan to build an experimental platform, which can weigh the absorbed water in a RH-controlled chamber as well as synchronously monitor the water gradient between top and bottom surface by powerful characterization methods (like time-resolved Raman). We hope the systematic results may be got soon.

To test the current output of MEG with inert electrode at 10% RH, the top electrode (Al electrode) was replaced by C electrode. A continuous I_{sc} output of about 350 μA is maintained over 4000s at 80% RH for MEG with C electrode as shown in Fig. R1-11a. The current output of MEG is still as high as 60 μA over 4000s at 10% RH in Fig. R1-11b. That means MEGs with C electrode can deliver a favorable current at low RH and a larger current at high RH, which are similar to the current output of MEGs with Al electrode. The reason why the MEG provides a favorable current at low humidity is mainly because the hybrid hydrogel based MEG possesses strong hygroscopicity and high ionization ability, allowing ion dissociation and migration in the MEG device even at low ambient humidity. [Advanced Materials. 2023, 35: 2300398] [Advanced Functional Materials. 2023; 33: 2211013] As verified in Fig. R1-10a, MEG can still absorb moisture from the air even at harsh environment and form water gradient conceivably by asymmetric absorption. Thus, the strong hygroscopicity and high ionization ability significantly promotes the current outputs of MEG even at low humidity.

Fig. R1-11 I_{sc} curve with the time for MEG with C electrode as top electrode at 80% RH (a) and 10% RH (b).

Moreover, we have added the reason why the MEG provides a high current at 10% in the revised manuscript on page 9 as follows: “It is notable that a favorable J_{sc} (ca. 150 $\mu\text{A cm}^{-2}$) is observed at 10% RH, derived from strong hygroscopicity of MEG to trigger ion diffusion even at low RH (Fig. S4b).” And we also added the moisture-absorption isotherm and the specific absorption kinetic curve into the supporting information on page 6 and Fig. S4.

6. The authors need to systematically compare this hybrid hydrogel with other reported moisture-generating materials in terms of moisture absorption as well as ion dissociation and ionic conductivity.

[Author's reply]: Thanks a lot for your suggestions. The moisture absorption capability of our MEG is about 68 wt% at 80% RH as shown in Fig. R1-9.

To calculate the ionic conductivity (δ_{dc}) of MEG device, electrochemical impedance spectrum (EIS) of MEGs at 80% RH was measured to get the internal resistance (R_i) (Fig. R1-12). Thus δ_{dc} can be deduced by the equation [ACS Applied Materials & Interfaces. 2017; 9: 11696]: $\delta_{dc} = \left(\frac{1}{R_i}\right)\left(\frac{t}{A}\right)$, where t is the hydrogel's thickness, and A is the hydrogel's surface area. Here R_i represents the overall internal resistance of MEG device. [Advanced Materials. 2016; 28: 1874] The calculated δ_{dc} for MEGs at 80% is ca. 1.02×10^{-3} S/cm.

Fig. R1-12 Nyquist plots for MEG with fitting plot at 80% RH and room temperature.

Based on the results of moisture absorption and ionic conductivity, we have compared our MEG with other published works as listed in the Table R1. Table R1 illuminates that our MEG owns medium ionic conductivity, decent moisture absorption capability, and especially superior power density as well as current density, compared to most of sustained MEGs under only moisture stimulation. As mentioned before, the power output of state-of-arts MEGs is mainly decided by water capturing ability and constructed water gradient. Here the hybrid hydrogel provides strong water affinity and the polymeric network to trap plenty of moisture nearby with asymmetric absorption structure, leading to a superior power density and current density. Overall, our MEG shows excellent electric performance compared to other sustained

MEGs.

Table R1. The performance comparison of current moisture electric generators.

Materials	Ionic conductivity (S cm ⁻¹)	Moisture uptake capability (wt%)	Current (μ A cm ⁻²)	Voltage (V)	Power (μ W cm ⁻²)	RH (%)	Ref.
PSSA/PDDA	$\sim 3 \times 10^{-3}$	30	4	1	5.52	75	2
Carbon black-sodium dodecyl sulfonate benzene	8×10^{-5}	150	7	0.7	226 μ W g ⁻¹	80	16
PSSA/R film	3.1×10^{-6}	109.9	160	0.82	88	50	17
PVA-PA-Gly gel	1.76×10^{-4}	30.0	240	0.8	35	80	18
G. sulfurreducens PCA film	1.6×10^{-3}	6.0	13	0.35	5.1	90	19
GO/PVA	6×10^{-7}	n.a.	92.8	0.85	n.a.	55	21
PSS/PVA textile	3.75×10^{-7}	n.a.	1.5	1	0.1	80	22
LiCl@ cellulon-Carbon black@ cellulon paper	1.5×10^{-6}	30	3	0.78	0.7	50	23
SA-SiO ₂ -RGO	1.1×10^{-3}	240	100	0.5	12	100	24
Waste activated sludge	1.8×10^{-3}	15.5	2.98	0.45	5.24	90	25
CS/SWNTs/PVA/CNF aerogel	2.6×10^{-3}	600	117	1.45	32.59	80	26
PSSA-kuromanin (chloride) film	0.1	227	300	0.8	n.a.	70	27
Sulfonate-polyaniline-bifunctionalized lignin	7.25×10^{-2}	40	0.125	0.28	44.5 W kg ⁻¹	99	28
Protein nanowire	1.2×10^{-6}	27	40	0.5	5	50	29
Asymmetric GO	3×10^{-6}	n.a.	0.6	0.45	2.02	25	30
PVA-AlgNa based hydrogel	1.02×10^{-3}	68	408	1.3	110	80	this work

Ion dissociation (ζ) is closely related to ionic conductivity. The more ion dissociation and

higher ion mobility, the greater ionic conductivity. [Nano Letters. 2023; 23: 5194][The journal of physical chemistry letters. 2019; 10:2313] The ion dissociation ξ is defined as free ions/total ions, which is described as the ratio of the molar conductivity measured by EIS (Λ_{imp}) over the molar conductivity estimated by the pulse-field-gradient spin-echo nuclear magnetic resonance (PG-NMR) obtained ionic self-diffusion coefficients and the Nernst-Einstein relation (Λ_{NMR}). [The Journal of Physical Chemistry B. 2006; 110: 19593] [The Journal of Physical Chemistry B. 2019; 123: 1348] Through EIS measurement, Λ_{imp} can be calculated as shown in Fig. R1-9. While Λ_{NMR} is hard to obtain at current stage since NMR lacks the accessory to construct a pulse-field-gradient field (PG) in our lab. In the future we will try to calculate the ξ by setting up the attachment PG for NMR. Another efficient predictor is to employ the pKa value, which is defined as the negative base-10 logarithm of the acid dissociation constant. [ACS Materials Letters. 2021; 3: 193] The lower pKa, the easier ion dissociation. Some previous works have reported about the pKa value of active materials like PSS, GO. [Advanced Materials. 2018; 30: 1705925] While most of literatures about MEGs nearly reported ξ or pKa value, making it hard to compare ξ with each other at current stage.

In addition, we have added EIS result and Table R1 in the supporting information of Fig. S13 on page 13 and Table S4 of page 23. The corresponding reference has been updated in the supporting information. Then the comparison result is discussed in the manuscript on page 16 as follows: “Combined with experimental results and DFT calculations, the synergistic effects of strong moisture-absorbing capability, long-standing water gradient, and then triggered abundant dissociated ion diffusion with enhanced ionic conductivity corporately empower the MEG outstanding electric generation, compared with other sustained MEGs (Table S4).”

Reviewer #2 (Remarks to the Author):

1. First and most important, though the power output performance reported in this paper is considerable, the potential influence of the Aluminum electrode corrosion is not clear. As the aluminum film was used as the top electrode, electrode corrosion due to the contact between water and metal is inevitable. It can also lead to a result of power generation. In fact, by using the electrode corrosion caused the Aluminum electrode, even the generator using a layer of water-soaked tissue as the conductive medium can produce a voltage output of 1.2 V. The electricity generated by the oxidation of metals with air belongs to the field of metal-air batteries, such as the Al-air batteries in the Power Sources 437 (2019) 226896. Besides, compared to other electricity generation methods due to the electrode corrosion, the power generation performance in this paper has no advantage.

Therefore, to reveal the influence of supramolecular hydrogel, the inert electrodes or carbon materials should be used instead of Al electrode in the power generation system, and the results of power generation should be provided and compared with previous studies. If the authors cannot provide the experimental results with inert or carbon electrodes, this paper should be rejected.

[Author's reply]: Thanks for your pertinent comments. To reveal the influence of supramolecular hydrogel, MEGs were prepared with carbon (C) electrode replacing Al electrode. The electricity generation performance was tested to compare with that of MEG using Al electrode as shown in Fig. R2-1.

The results in Fig. R2-1a show that MEGs with C electrode yield favorable electric output, which are comparable to MEGs with Al electrode. A stable V_{oc} of ca. 1.2 V is observed for MEG with C electrode (Fig. R2-1b), which approaches that of MEG using Al electrode. Besides, a continuous I_{sc} output of about 350 μ A is maintained over 4000s at 80% RH as shown in Fig. R2-1c, which also catches up with that of MEG using Al electrode. Such small difference for MEGs with different electrodes is acceptable and also observed in the other references. [Nature Nanotechnology. 2021; 16: 811] [Energy & Environmental Science. 2023; 16: 2338] [Energy & Environmental Science. 2019; 12: 972] Based on above comparison results, it can be safely

concluded that the supramolecular hydrogel is the key for electric generation for the MEG instead of electrode. The supramolecular hydrogel enhances the moisture absorption of MEG to promote the sufficient chemical conversion energy, meanwhile provides hydrophilic polymeric network to trap plenty of moisture nearby with asymmetric absorption structure, consequently inducing high power output. Similar with other previous works, Al electrode is actually a common selection for MEGs. [Energy & Environmental Science. 2016; 9: 912] [Energy & Environmental Science. 2023; 16: 2338] Compared to the inert electrodes like C, Al is highly flexible, lightweight, easily accessible and fairly cheap, which is desirable for scalable and low-cost MEGs towards wide applications.

Fig. R2-1 The electric generation performance of the MEGs with C-Al and C-C electrodes. (a) Comparison of electric outputs of MEGs with C-Al and C-C electrodes. (b) The V_{oc} curves of MEGs with C-Al and C-C electrodes, respectively. (c) The I_{sc} curves of the MEGs with C-Al and C-C electrodes, respectively.

Furthermore, MEGs with C-C electrode were prepared to measure the humidity and temperature-dependent characteristics. Fig. R2-2a shows that the average I_{sc} augments monotonically from ca. 50 μ A up to 330 μ A with the evolution of RH from 10 to 80%. The accelerated current mainly derives from increased moisture absorption to trigger substantial ion diffusion at high humidity. Meanwhile, V_{oc} presents a gradual rise to ~ 1.2 V with the RH. In addition, our MEG also shows great adaptability towards a wide range of temperature. From 0 to 50 $^{\circ}$ C, the average J_{sc} shows a sharp climb from about 15 μ A to 552 μ A (Fig. R2-2b). Such obvious growth benefits from the synergistic effect of accelerated ion transport rate and ion concentration at high temperature. The elevated temperature also enhances V_{oc} to about 1.2 V. The humidity and temperature-dependent characteristics of MEGs with C-C electrode display the same tendency with that of MEGs with C-Al electrode. Overall, MEGs with C-C electrode

demonstrate comparable electric output to MEG with C-Al electrode, even changing humidity/temperature conditions.

Fig. R2-2 The I_{sc} and V_{oc} of MEG with C-C electrodes at different RHs tested at room temperature (a) and at different temperatures tested at 80% RH (b).

To further verify the energy generated from the moisture, we tested the electric output of MEG in a N_2 environment with moisture. Since oxygen is insulated in this way, the possible influence like electrode oxidation is excluded effectively. Thus the generated electricity mainly depends on the chemical conversion energy of absorbed moisture. Fig. R2-3 shows that MEG in an oxygen-insulated environment generates a continuous V_{oc} of about 1.3 V and a high I_{sc} of ca. 400 μA at 80% RH, comparable to MEG tested in ambient air. Combined with above results, it is reasonable to say that the electricity generation in our MEG device mainly derives from moisture absorption by supramolecular hydrogel.

Fig. R2-3 The V_{oc} (a) and I_{sc} (b) output of MEG tested at 80 % RH in a N_2 environment.

In addition, the electric generation performance of MEGs assembled by inert electrodes at

different RHs and temperature, as well as in N₂ environment has been added in the supporting information of Fig. S7 and S8 on page 9-10. Besides, we also added the discussion in the manuscript on page 9 as follows: “Furthermore, MEGs with different top electrodes output favorable electric performance and display similar humidity/temperature-dependent characteristics in Fig. S7, suggesting electricity generation mainly derives from moisture absorption by supramolecular hydrogel. The comparable V_{oc} and I_{sc} outputs are also found in an oxygen-insulated environment with 80% RH, further excluding the influence of electrode (Fig. S8). [Journal of Power Sources. 2019; 437: 226896.]”

2. As mentioned in the results and discussions, "As a natural polysaccharide, the polyanionic AlgNa features with numerous hydroxyl groups," "supramolecular AlgNa/Ca ionically crosslinked network is formed with abundant carboxyl functional groups (e.g., -COONa and -COOCa) in Fig. 1a." The FTIR spectra of AlgNa and polyvinyl alcohol (PVA)-sodium alginate (AlgNa) are suggested to be provided.

[Author’s reply]: Thanks a lot for your valuable suggestions. As shown in Fig. R2-4, FTIR spectrum of pure AlgNa shows the bands around 3277, 1608, and 1409 cm⁻¹, corresponding to the stretching of -OH, -COO⁻ (asymmetric), and -COO⁻ (symmetric) group, respectively. In contrast, an obvious shift of -COO⁻ stretching band to higher wavenumber 1654 cm⁻¹ is observed in PVA-AlgNa based hydrogel, indicating the crosslinking Ca²⁺ with -COO⁻ chains of AlgNa. [ACS applied materials & interfaces. 2020; 12: 23474]

The FTIR result has been added in the supporting information on page 5 in Fig. S1. And the discussion is also added in the manuscript on page 5 as follows: “As a natural polysaccharide, the polyanionic AlgNa features with numerous hydroxyl groups (Fig. S1), thus presenting prominent water-affinity feature, which is expected to enhance the water absorption of MEG. Moreover, by adding crosslinker of CaCl₂, an obvious blue shift of -COO⁻ stretching band of PVA-AlgNa in FTIR spectrum is observed, suggesting the crosslinking Ca²⁺ with -COO⁻ of AlgNa.²⁵”

Fig. R2-4 FTIR spectra of pure AlgNa hydrogel and PVA-AlgNa hydrogel.

3. The moisture-absorption isotherm of the AlgNa and polyvinyl alcohol (PVA)-sodium alginate (AlgNa) are suggested to be provided, to evaluate the electrical energy output performance under different humidity conditions.

[Author's reply]: Many thanks for your useful comments. The quantified water absorption capability of PVA-AlgNa based hydrogel in our work and pure AlgNa hydrogel at various RHs, namely the water absorption/desorption isotherm, was identified by a dynamic vapor sorption analyzer (BSD-DVS) at a constant temperature of 25 °C. Hygroscopic behaviors of PVA-AlgNa hydrogel and AlgNa hydrogel are shown in Fig. R2-5. Both PVA-AlgNa and AlgNa hydrogels show a gradual increment of water uptake with the RH (Fig. R2-5a and b). More importantly, the moisture uptake capability of PVA-AlgNa hydrogel is obviously higher than that of AlgNa hydrogel at each RH. At a low RH of 10%, the moisture uptake capability of PVA-AlgNa based hydrogel (2.1 wt%) is ten times larger than that of AlgNa hydrogel, which shows strong moisture absorption of PVA-AlgNa even at harsh environment. At a high RH of 90%, an exceedingly high moisture uptake capacity (ca. 90 wt%) of PVA-AlgNa hydrogel is observed, which is twice than that of AlgNa at the same condition (90% RH) (Fig. R2-5b).

Fig. R2-5 Moisture-absorption isotherm at different RHs of PVA-AlgNa based hydrogel (a) and AlgNa hydrogel (b), respectively. The test condition is at 25 °C.

Furthermore, Fig. R2-6 depicts the relationship of the current performance and moisture uptake capability with the RH variation. With the evolution of RH from 10 to 80%, the average current augments monotonically up to 407 μA along with a gradual enhanced moisture uptake capability. The simultaneous escalated current and moisture uptake hints the essential role of moisture in power generation. A slight decrease of current is observed at 90% RH possibly due to reduced water gradient with excessive moisture absorption.

Fig. R2-6 Moisture-absorption isotherm and current output of PVA-AlgNa based hydrogel versus RH variation.

Moreover, Fig. R2-6 about moisture-absorption isotherm and current output of MEG versus RH variation has been added in the supporting information of Fig. S4 on page 6. And the discussion has been added in the manuscript on page 8 as follows: “With the increase of RH from 10 to 80%, the average J_{sc} augments monotonically up to 407 $\mu\text{A cm}^{-2}$, well corresponding to enhanced moisture uptake capability of MEG (Fig. S4a). The escalated J_{sc} and moisture

uptake capability hint the essential role of moisture in power generation.” The manuscript on page 10 is modified into: “The increased RH can enhance water absorption capability (Fig. S4). The current in Fig. 3a also represents a positive boost after changing RH from 5% to 75%.”

4. To better explore the influence of environmental humidity on the power generation performance, humidity response should be tested. The instantaneous change in ambient humidity should be introduced to observe charging of performance outputs.

[Author’s reply]: Thanks for your constructive comments. To quantitatively measure the prompt response to the environmental humidity, the current output of MEG is synchronically tested with the cycling variation of RH. The result is shown in Fig. R2-7. When RH decreases from 60% to 30%, the short-circuit current (I_{sc}) rapidly drops from ca. 245 μA to ca. 145 μA . Then the I_{sc} remains for 1 min with RH holding at 30%. Subsequently, I_{sc} immediately rises up accompanied by fast hydration. The regular change of current output of MEG is well observed when response to the cycling variation of RH, suggesting the power generation is closely related to the hydration process.

The detailed discussion has been added in the supporting information of Fig. S11 on page 12. The manuscript on page 10 was revised to: “With the intermittent and periodic RH variation from 30% to 60%, a regular change in current output of MEG synchronously responses (Fig. S11).”

Fig. R2-7 Current cycles of one MEG in response to the intermittent and periodic RH variation from 30% to 60%.

5. As mentioned in the Working mechanism of MEG, “In addition, the current is immediately promoted after relieving the vacuum” Fig. 3a shows an extremely low current for MEG at low RH of 5%, mainly due to few ion dissociation. The current gradually increases after exposing MEG at 75% RH with abundant ions diffusion.” Is the main effect of the increase in ambient humidity to reduce the internal resistance of the device or to trigger plentiful ions diffusion? The impact of environmental humidity variations on voltage output should be discussed in detail, not only focusing on the changes in current.

[Author’s reply]: Thanks a lot for your useful suggestions. Firstly, it is believed that the increase of RH is beneficial to ion diffusion, which is enabled by water absorption. [Energy & Environmental Science. 2019; 12: 1848][Energy & Environmental Science. 2019; 12: 972] The ion diffusion for MEG exposed in the air has been proved by EDS in Fig. R2-8, which clearly demonstrates that Ca^{2+} ions transport from top to bottom surface during the charging process by moisture absorption.

Fig. R2-8 The chemical component characterization of detective ion (Ca^{2+}) variation between the top and bottom surface of MEG in different states by EDS.

To investigate the effect of RH on the internal resistance (R_i), electrochemical impedance spectra (EIS) of MEGs at 80% and 30% RH were measured to calculate the ionic conductivity (δ_{dc}), deduced by the equation [ACS Applied Materials & Interfaces. 2017; 9: 11696]: $\delta_{dc} = \left(\frac{1}{R_i}\right)\left(\frac{t}{A}\right)$, where t is the hydrogel’s thickness, and A is the hydrogel’s surface area. Here R_i represents the overall internal resistance of MEG device. [Advanced Materials. 2016; 28: 1874]

Fig. R2-9a and b display the simulation results of Nyquist plots of MEGs at 80% and 30% RH, respectively. The corresponding δ_{ac} values for MEG devices are 1.02×10^{-3} S/cm at 80% RH and 7.46×10^{-5} S/cm at 30% RH, respectively. That means the ionic conductivity of MEG enhances with the RH increase. Furthermore, a decreased slope for MEG at 30% RH indicates that the charging process was limited by ion diffusion in the hydrogel. [The Journal of Physical Chemistry C. 2018; 122: 194] From both above results, it is inferred that the increase in ambient humidity can synergistically reduce the internal resistance of MEG device and trigger the ion diffusion, resulting in a soaring current output. Similar phenomenon was also reported in previous work [Energy & Environmental Science. 2019; 12: 1848], where by increasing the RH from 5% to 95%, the ionic conductivity of the GO assembly showed a two-orders-of-magnitude enhancement with promoted ion transport.

The EIS results are added in the supporting information of Fig. S13 on Page 13-14 and are discussed in the manuscript on page 11 and 16 as follows: “The severalfold enhanced moisture intake capability not only enables MEG with plentiful ions dissociation but also triggers ion diffusion as well as enhances its ionic conductivity as shown in Fig. S13, thus greatly escalating the generation of current output.” “Combined with experimental results and DFT calculations, the synergistic effects of strong moisture-absorbing capability, long-standing water gradient, and then triggered abundant dissociated ion diffusion with enhanced ionic conductivity corporately empower the MEG outstanding electric generation.” To better clarify the working mechanism, the manuscript on page 10 is modified into “The increased RH can enhance water absorption capability (Fig. S4). The current in Fig. 3a represents a positive boost after changing RH from 5% to 75%. With the intermittent and periodic RH variation from 30% to 60%, a regular change in current output of MEG synchronously responses (Fig. S11).”

Fig. R2-9 The Nyquist plots of MEGs at (a) 80% RH and (b) 30% RH with fitting plots.

Secondly, the voltage output versus RH is depicted in Fig. S5a and discussed on page 7 of supporting information as follows: “Fig. S5a shows that the voltage outputs are gradually improved from ca. 0.8 V to 1.3 V with increasing RH to 80%. As we know, the voltage output is closely related to water gradient, or an ion concentration gradient. It is proposed that the increased RH promotes more ions dissociation and forms a larger ion concentration difference, positively enhancing voltage output. When the RH reaches 90%, the voltage slightly decreases, probably due to the inferior moisture gradient by saturation. Similar trend is observed for voltage change with the rising of temperature (Fig S5b), mainly deriving from triggered ion dissociation and diffusion. The rising of temperature promotes ion dissociation to improve ion concentration difference meanwhile accelerates the ion diffusion rate, boosting a high V_{oc} of about 1.3 V and a large current. At a low temperature of -25 °C, there still exists a voltage of 0.9 V, demonstrating a wide environmental adaptability of MEG.”

The voltage changes in the manuscript on page 9 are also modified into: “Besides, V_{oc} presents a gradual rise from ca. 0.8 to 1.3 V with RH rising to 80% (Fig. S5a), due to a larger ion concentration difference formed at an improved RH.” And “The elevated temperature also improves V_{oc} from 0.9 V slowly until reaching a platform of ca. 1.3 V (Fig. S5b), mainly due to enhanced ion concentration difference at high temperature.”

6. The DFT calculation found that the interaction strength between H₂O and polymers decreases with the order: AlgCa > AlgNa > PVA. And, the power output of MEG is mainly decided by the constructed water gradient. Therefore, the experimental validation should be confirmed under the same ambient humidity and temperature that the sustained durability in electrical energy output as follows: AlgCa > AlgNa > PVA.

[Author’s reply]: Thanks a lot for your critical comments. To verify the sustained durability in power output for AlgCa > AlgNa > PVA experimentally, we directly prepared MEGs with water-soluble PVA hydrogel and AlgNa hydrogel, separately. While for AlgCa, it is insoluble in water and cannot be directly prepared. [Polymer international. 2008; 57: 171] As an

alternative to investigate the effect of AlgCa, PVA-AlgNa based hydrogel in our work was applied, where AlgCa ionically cross-linked network is formed as shown in Fig. R2-4. The sustained voltage and current are compared as shown in Fig. R2-10.

Fig. R2-10 (a, b, c) The V_{oc} curves of MEG devices based on PVA hydrogel, AlgNa hydrogel, PVA-AlgNa based hydrogel over time under the same environment and room temperature, respectively. (d, e, f) The current output of MEG devices based on PVA hydrogel, AlgNa hydrogel, PVA-AlgNa based hydrogel with 1 k Ω external resistor under the evolution of time at 70% RH and room temperature, respectively. The insert shows the circuit diagram of MEG with one resistor.

For the voltage output, PVA based MEG exports a stable V_{oc} of about 0.8 V after a drop from 1.0 V within 20 minutes (Fig. R2-10a). As for AlgNa based MEG, a V_{oc} of about 1.0 V is sustained for more than 60 hours (Fig. R2-10b). In contrast, PVA-AlgNa based MEG can generate a continuous DC V_{oc} of about 1.3 V for more than 90 hours (Fig. R2-10c). That means the sustained durability of voltage output follows the order: AlgCa > AlgNa > PVA.

The sustained durability in current was also tested for these three MEGs by loading with 1 k Ω external resistor. The current of PVA based MEG drops by 50% within 2 h (Fig. R2-10d). While the current of AlgNa based MEG reduces by 33% at the same time. In clear contrast, PVA-AlgNa based MEG presents a steady current output for about 17 h (Fig. R2-10f). These results further demonstrate that the sustained durability in electrical energy output follows the order:

AlgCa > AlgNa > PVA, which is in accordance with the DFT simulation results.

In addition, experimental validation about the sustained durability in electrical energy output has been added and discussed in the supporting information of Fig. S15 on page 15-16. Besides, the manuscript on page 13 was revised into: “In Fig. 3d-f, it clearly reveals that the interaction strength between H₂O and polymers decreases with the order: AlgCa > AlgNa > PVA, which is well verified by the sustained durability of electrical outputs in Fig. S15.”

7. In Fig. 6, the application demonstration, more information about the energy management circuit should be given. How to maintain the stable voltage for the continuous work of the lamp bulb? How long does the generator array take (charging process) to accumulate the required the voltage level for the bulb?

[Author’s reply]: Thanks a lot for your comments. Fig. 6d shows that a lamp bulb is illuminated continuously by a large-scale MEGs bank with a 10*24 in parallel*serial combination (Video S2). It is worth noting that the lamp bulb is directly lighted by integrated MEG arrays without any voltage converter or energy management at current stage. That means the predesigned MEG arrays can render sufficient power output to drive the bulb, which also well verifies the great scalability and high electricity generation of MEG. In addition to that, we also employ integrated MEG arrays to directly power other commercial electronics (like smart watch, LCD clock) without auxiliary energy-storage devices and rectifying circuits as shown in Fig. 6.

It is believed that power management circuit is of use to realize efficient energy harvesting meanwhile provide stable voltage output. [IEEE journal of solid-state circuits. 2009; 44: 2824] [Advanced Science. 2020; 7: 2001362] However, energy management has been barely explored for the new-emerging MEG field until now. In the future work, we will try to explore power management system for MEG, leading an advanced guidance for optimized energy output in an optimized experiment conditions. By this way, we can figure out appropriate methods and the time needed to accumulate the power efficiently and maintain stable output voltage.

In addition, we have modified the manuscript about the power management system on page 19:

“The predesigned MEG banks can render sufficient power output to drive optoelectronics for daily use. Beyond that, the MEG bank is small, lightweight, and flexible with sustained power output, promising the practical applications for IoTs. From the perspective of practical application, it is better to develop the power management system of the large-scale MEG bank for efficient energy harvesting and output in the future.”

Reviewer #3 (Remarks to the Author):

The manuscript reported a high-performance moisture-electric generator based on PVA-sodium alginate supramolecular hydrogel. The reported current and power outputs of the single unit MEG reach 1.31 mA cm^{-2} and $110 \text{ } \mu\text{W cm}^{-2}$ with the open-circuit voltage of 1.3 V. These performance metrics exceed most of the existing literature by more than one order of magnitude. Such DC current keeps stable for more than 17 hours. The authors attributed the excellent performance of their MEG to enhanced moisture absorption and slow water diffusion. Towards practical applications, the authors demonstrated the highest short circuit of 65 mA through a parallel-connected MEG array with 240 units. A 2-serial MEG bank was able to power an LCD clock for one month.

Given the impressive short circuit current, long-term steady electrical output, and scalability of MEG, I think the PVA-alginate MEG will have a broad impact and attract readers from various fields. However, the authors did not provide enough convincing evidence to support the claim of slow water diffusion inside the PVA-alginate hydrogel. Therefore, I think significant revision is needed and the following concerns need to be resolved before this manuscript can be published in nature communication.

[Author's reply]: Thanks very much for your encouraging comments.

1. I think more experimental results or calculation results are needed to support the claim of slow water diffusion. The authors should clearly elaborate what is the water diffusion time in their MEG and how this value compares to previously reported humidity-driven electric generators.

[Author's reply]: Thanks a lot for your pertinent comments. In our work, “slow diffusion” means that ion water clusters transport slower than other states of water (like weak bound water, strong bound water), which is decoupled from 2D-FTIR spectroscopy (Fig. 4e and f). By decoupling overlapped bands of O-H stretching band, the specific sequence orders of bands are $3540 > 3115 > 3304 \text{ cm}^{-1}$. That means water diffusion follows the sequence: weak bound water (3540 cm^{-1}), strong bound water (3115 cm^{-1}), cluster water (3304 cm^{-1}).

To better elaborate slow water diffusion, we further calculate the water diffusion coefficient to assess the water diffusion speed. The quantitative information is extracted from the 1D-FTIR profiles by integrating the intensities of the 2D-FTIR patterns. Hence, the intensities of three decoupled peaks (3540, 3115, 3304 cm^{-1}) with the time were obtained. Furthermore, equation 1 is given to estimate the effective diffusion coefficient of water from FTIR spectra based on the Fickian diffusion. [The Journal of Physical Chemistry B. 2008; 112: 2880]

$$\frac{A_t}{A_\infty} = 1 - \frac{8\gamma}{\pi[1 - \exp(-2\gamma L)]} \times \sum_{n=0}^{\infty} \left\{ \frac{\exp(g) [\exp(-2\gamma L) + (-1)^n (2\gamma)]}{(2n + 1)(4\gamma^2 + f^2)} \right\} \quad (1)$$

Where

$$g = \frac{-D(2n + 1)^2 \pi^2 t}{4L^2}, f = \frac{(2n + 1)\pi}{2L}$$

In equation 1, A_t is the band absorbance of the FTIR spectra at time t , A_∞ is the band absorbance at equilibrium, γ is the penetration depth of the evanescent wave, L is the thickness of the polymer membrane (invariable), and D is the diffusion coefficient. The D values of different states of water can be calculated by a nonlinear curve fitting to equation 1 from the variation of the three decoupled peaks (3540, 3115, 3304 cm^{-1}) versus time. [Macromolecules. 2002; 35: 5500]

The results are shown in Fig. R3-1. The D values for three peaks also follow the order: 3540 > 3115 > 3304 cm^{-1} . That means weak bound water diffuses in the fastest way, while ion water cluster diffuses in the slowest way due to intense attraction force by AlgCa/Na network. The smaller D value of ion water cluster well elaborates the viewpoint of “slow diffusion”. To clarify it better, “slow diffusion” is modified as “relatively slow diffusion”. Since the water diffusion time would be affected by many factors, such as temperature, RH, the thickness of hydrogel, it is more reasonable to describe the water diffusion time by D value. After reading lots of literatures about MEGs, we find there exists few works reporting about D value or diffusion time, making it difficult to compare to previously reported MEGs. More importantly, slow diffusion in our work means that ion water cluster is slower than other states of water.

Therefore, it makes sense only when D value of ion water cluster compares with that of other states of water for our own MEG.

Fig. R3-1 The integrated intensities of the three decoupled peaks versus time, (a) 3540 cm^{-1} ; (b) 3115 cm^{-1} , (c) 3304 cm^{-1} .

We have modified the manuscript on page 4, 13, 15 about “slow diffusion” as “relatively slow diffusion”. Besides, the quantified water diffusion coefficients have been added in the supporting information on page 18 in Fig. S20. We also add the discussion in the manuscript on page 14 as follows: “According to the Noda rules, the specific sequence orders of bands mentioned above are $3540 > 3115 > 3304\text{ cm}^{-1}$, which agrees well with the corresponding water diffusion coefficients calculated by Fickian diffusion equation [The Journal of Physical Chemistry B. 2008; 112: 2880] as shown in Fig. S20.....The smaller water diffusion coefficient of ion water clusters well verifies the relatively slow diffusion compared to that of other states of water. Therefore, MEG presents a unique behavior of “fast absorption” of moisture and “relatively slow diffusion” of ion water clusters.”

2. I assume that the hydrogel conducts current through ion transport in the short-circuit condition. In this process, which ion dominates the process (e.g., Na^+ , Ca^{2+} , Cl^- or the dissociated H^+ , OH^- ion from the water)? The ion transport is expected to establish an equilibrium in the short circuit condition. What is the time scale of such equilibrium in the MEG and will this ion equilibrium discourage the current output of MEG? The authors should provide some discussion on these points in their manuscript.

[Author’s reply]: Thanks for your useful comments. We agree with you the viewpoint that the

ion release and transport was conducted through the hydrogel, then inducing an electric output in the external circuit. [Energy & Environmental Science. 2019; 12: 1848] By short-circuit treatment, the transport of Na^+ , Ca^{2+} , Cl^- ions has been observed according to the EDS results as shown in Fig. R3-2. The discharge and recharge processes of MEG go through short-circuit treatment. A reversed $\Delta c(\text{Ca})$ from negative into positive is observed, suggesting an obvious transport process. Similar transports are observed for Na and Cl ions. In contrast, water dissociation is supposed to make little contribution to the ion conductivity in such PVA-AlgNa hybrid hydrogel system due to larger dissociation energy. [Journal of Membrane Science. 1999; 162: 145] Thus the transport of Na^+ , Ca^{2+} , Cl^- ions dominates the process.

Fig. R3-2 The chemical component characterization of detective ion variation between the top and bottom surface of MEG in different states by EDS. (a) Ca ion. (b) Cl ion. (c) Na ion.

Also, it is expected that the ion transport tends to establish an equilibrium in the short circuit condition. As shown in Fig. R3-3, when the MEG goes through short-circuit treatment of 15 days, the ion concentration difference between top and bottom surface becomes much smaller, compared to that of MEG with 5 hours' short-circuit treatment. That means the ion diffusion is closer to equilibrium state after 15 days' short-circuit treatment. When testing the short-circuit current (I_{sc}) of MEG continuously, Fig. R3-4 shows that I_{sc} gradually decays over a long time, which may derive from the saturation of water absorption, leading to a reduced ion

concentration difference. Despite that, the inserts in Fig. R3-4 show the current still persists for >120 h, indicating that new energy conversion still goes on accompanied by dynamic water adsorption-desorption exchange at the interface. So, it can be learned that the timescale to establish the ion equilibrium is more than hundreds of hours.

Fig. R3-3 The ion distribution between top and bottom surface for MEG suffering from 5 hours' and 15 days' short-circuit treatment. (a) Ca element. (b) Cl element. (c) Na element.

Fig. R3-4 (a) The I_{sc} output of an MEG device with the evolution of time. (b, c, d) The inserts show the current curve of the MEG device at different time slots.

Combined with the EDS results, we added the discussion in the manuscript on page 15 as

follows: “It is worth noting that the absolute value of $\Delta c(\text{Ca})$ becomes larger, which suggests that moisture enables extra dissociated Ca^{2+} ions release from polymer chains ($-\text{COOCa}$) and then triggers these Ca^{2+} ions migration..... Except for the dissociated ions, free ions like Cl^- , Na^+ also presents a similar transport process (Fig. S21). When extending the short-circuit time, Δc values become smaller (Fig. S22) and ion transport tends to establish an equilibrium. Meanwhile, the dissociated positive ions drift back to their initial place driven by the electrostatic attraction of bulky immovable negative Alg^- chains. And the free positive ions tend to reunite the free Cl^- ions. Nonetheless, an ion-diffusion-induced current still outputs over hundreds of hours as shown in Fig. S14b.”

3. What is the error bar in Fig. 4g? I doubt the accuracy of using EDS to characterize the ion variation on the surface of the MEG. The observed variation is below 1%. However, to use EDS, the sample needs to be vacuumed in an SEM chamber, which I suppose will significantly influence the ion distribution on the sample surface. What is the area size of the EDS measurement to determine the Ca variation? Additionally, did the author measure the variation of other ions on the MEG surface using EDS?

[Author’s reply]: Thanks a lot for your comments. To improve the accuracy of using EDS to characterize the ion variation, we retested 4-6 positions on the surface for each sample and calculate the average value as well as deviation. The area size of EDS mapping is about $200 \times 262 \mu\text{m}^2$. The EDS mapping results with error bar about Ca^{2+} , Na^+ , Cl^- ions are shown in Fig. R3-2, which revalidates similar ion diffusion trend during the charging and discharging process as before. Here $\Delta c(\text{ion})$ is defined as the difference of ion content between the top and bottom surfaces of MEG. Fig. R3-2 show that $\Delta c(\text{Ca})$ and $\Delta c(\text{Cl})$ are about or more than 1% with small error bar in charge-discharge process. That means all 4-6 points for each sample show similar ion diffusion trend. Thus it is believed that the EDS results are reasonable.

Fig. R3-2 The chemical component characterization of detective ion variation between the top and bottom surface of MEG in different states by EDS. (a) Ca element. (b) Cl element. (c) Na element.

To evaluate the influence of vacuum on the ion distribution during EDS testing, we in situ detect the ion distribution of six spots on the surface with the vacuum time. Fig. R3-5 shows the ion concentration of each spot through vacuuming at 0.25 hours and 3 hours. It clearly demonstrates a very slight change on the ion distribution during the vacuum process no matter of Ca^{2+} , Na^+ , Cl^- ions. Besides, the average values keep almost the same. Thus, it is safe to say that vacuum exerts little influence on the ion distribution.

Fig. R3-5 In situ characterization of ion distribution on bottom surface of MEG device by vacuuming 0.25 hours and 3 hours. The MEG device goes through short circuit treatment by 15 days. (a) Ca element. (b) Cl element. (c) Na element. Every data point indicates an independent testing at different places of the sample.

In addition, Fig. 4g in the manuscript on page 13 has been modified with the updated EDS results. And EDS results for Na⁺ and Cl⁻ ions have been shown in Fig. S21 in the supporting information on page 19. The discussion about ions variation in the manuscript on page 15 is modified as before.

4. In addition to my previous question, could the authors provide information on how the ion distribution on the surface of the MEG changes after it has been subjected to a steady short current condition for an extended period of time, for example, 1 week or 1 month?

[Author's reply]: Thanks a lot for your comments. Fig. R3-3 shows the change of ion distribution on the top and bottom surfaces of the MEG after the short-circuit treatment for 15 days. The Ca content difference between top and bottom surface becomes smaller compared to that of MEG suffering from 5 hours' short-circuit treatment (Fig. R3-3a). The reason lies in

that the short-circuit treatment promotes the ion diffusion closer to equilibrium state. The similar phenomenon is observed for Cl^- ion in Fig. R3-3b. The larger deviation makes less obvious change for Na^+ ion.

Fig. R3-3 The ion concentration difference between top and bottom surface for MEG suffering from 5 hours' and 15 days' short-circuit treatment. (a) Ca element. (b) Cl element. (c) Na element.

In addition, the ion distributions are added in the supporting information of Fig. S22 on page 20. Besides, the manuscript on page 15 is modified as follows: “When extending the short-circuit time, Δc values become smaller (Fig. S22) and ion transport tends to establish an equilibrium. Meanwhile, the dissociated positive ions drift back to their initial place driven by the electrostatic attraction of bulky immovable negative Alg^- chains. And the free positive ions tend to reunite the free Cl^- ions.”

5. What is the water concentration by weight percent when the MEG obtains the best energy conversion efficiency? How long does it need to obtain the optimum performance from a fully dried state? I assume that a fully dried MEG will have a very low short circuit current because of the dominance of ionic conductivity in the hydrogel. I think the internal resistance of the

device in the dry and water-absorb states should also be provided.

[Author's reply]: Many thanks for your comments. For the energy conversion efficiency, it can be calculated based on the Gibbs free energy in principle. [Advanced Materials. 2023; 2209661] The overall energy input ΔG_{in} is defined as

$$\Delta G_{in} = \mu_{gas} - \mu_{ads} \quad (1),$$

where μ_{gas} and μ_{ads} represent the chemical potential of the moisture in the gaseous and adsorbed state. ΔG_{in} is related to chemical potential conversion of the total absorbed moisture. The electric output W is defined as

$$W = \int U(t) \cdot I(t) dt \quad (2),$$

where $U(t)$ and $I(t)$ are the open-circuit voltage output and short-circuit current output of the MEG. The energy conversion efficiency η , thus, is calculated by

$$\eta = \frac{W}{\Delta G_{in}} \quad (3),$$

Fig. R3-6a clearly demonstrates the correlation between water concentration and electricity generation. Moisture absorption of MEG includes two stages: the first one is linear absorption quickly, the second one is reaching the saturation state with moisture adsorption and desorption dynamically. ΔG_{in} is expected to evenly increase at the first stage. Meanwhile, I_{sc} maintains at hundreds of μA level despite accompanying with a slight decrease (Fig. R3-6b). And V_{oc} keeps steady (Fig. R3-6f), which means that W will slightly lower down at the first stage. Thus the maximum η is estimated to be at the beginning and then η will decrease with the time at the first stage. It is mainly because the water gradient is constructed at first and gradually decays with the water diffusion. At the saturation state, the increase of ΔG_{in} becomes very slow when the moisture adsorption and desorption reach equilibrium at the interface. Besides, the I_{sc} drops to a very low level (Fig. R3-6d and e) despite accompanying with a steady V_{oc} . Therefore, η is predicted to be low at saturation stage. Based on the above discussion, energy conversion efficiency is high during the quick moisture uptake stage, and then becomes lower when achieving saturation. From the perspective of practical use, it would be more meaningful to use

MEG at high η value range.

Fig. R3-6 Moisture uptake capability and current output of MEG versus time synchronously. (a) The mass increase of MEG by moisture adsorption versus time under 80% RH and room temperature. (b) The concurrent I_{sc} output of MEG versus time at the same condition. (c-d) The tested current output of MEG at different time slot. (f) The V_{oc} output of MEG versus time at the same condition.

The ionic conductivity (δ_{dc}) or internal resistance (R_i) of MEG is calculated by electrochemical impedance spectroscopy (EIS) result as shown in Fig. R3-7. To prepare MEGs in the dry and water-absorbed states, the fabricated MEGs was dried at 20 % RH for one week and was put in 80% RH for over 24 hours, respectively. Herein, R_i represents the overall internal resistance of MEG device, which includes the resistance of hydrogel and the interfacial resistance between hydrogel and electrodes. [Advanced Materials. 2016, 28, 1874] The experimental impedance data are simulated to calculate the ionic conductivity by the equation [ACS Applied Materials & Interfaces. 2017; 9: 11696]: $\delta_{dc} = \left(\frac{1}{R_i}\right)\left(\frac{t}{A}\right)$, where t is the hydrogel's thickness, and A is the hydrogel's surface area. The calculated δ_{dc} for MEG in water-absorbed state is ca. $1.57 \cdot 10^{-3}$ S/cm, while that of MEG in dried state is about $3.65 \cdot 10^{-5}$ S/cm. That means the

ionic conductivity of the MEG shows a two-orders-of-magnitude enhancement under water-absorbed state. Furthermore, a decreased slope for MEG in a dried state indicates that the charging process was limited by ion diffusion in the hydrogel. [The Journal of Physical Chemistry C. 2018; 122: 194] That means MEG in a dried state displays a large internal resistance and retarded ion diffusion process, thus delivering a low short-circuit current. [Electrochimica Acta. 2016; 201: 251] [Advanced Functional Materials. 2007; 17: 2645] [Advanced Materials. 2017; 29: 1700974]

Fig. R3-7 The Nyquist plots of MEG in the water-absorb (a) and fully-dried states (b).

In addition, the ionic conductivities of MEG device at 30% and 80% RH were added in the supporting information on Page 13-14 and Fig. S13, which also well elaborates the internal resistance of the device at different hydration states. We also discussed it in the manuscript on page 11 and 16 as follows: “The severalfold enhanced moisture intake capability not only enables MEG with plentiful ions dissociation but also triggers ion diffusion as well as enhances its ionic conductivity as shown in Fig. S13, thus greatly escalating the generation of current output.” “Combined with experimental results and DFT calculations, the synergistic effects of strong moisture-absorbing capability, long-standing water gradient, and then triggered abundant dissociated ion diffusion with enhanced ionic conductivity corporately empower the MEG outstanding electric generation, compared with other sustained MEGs (table S4).”

6. What is the recharging time for the MEG unit? The authors have demonstrated a steady

current output of 0.25 mA for 17 hours. Can current output at the level of 250 μA be steady for a month without recharging?

[Author's reply]: Thanks a lot for your kind comments. The current will decrease gradually with the time after a steady output of 0.25 mA for 17 hours as shown in Fig. R3-8, which may derive from the saturation of water adsorption over time. After saturation, water adsorption at a solid surface is a dynamic equilibrium involving constant adsorption-desorption exchange at the interface. [Nature. 2020; 578: 550] Since the continuous adsorption of water molecules couples with the ion dissociation process in the MEG, the ion concentration is expected to establish an equilibrium over a long period. [Energy & Environmental Science. 2019; 12: 1848] Thus a deteriorated current is observed with the pass of time. Despite that, the insert in Fig. R3-8 shows the current can still output 8 μA after 1 month, indicating that new energy conversion still goes on accompanied by dynamic water adsorption-desorption exchange at the interface.

Fig. R3-8 The current curve of MEG over 1 month with the loading resistor of 1 $\text{k}\Omega$ under 70% RH. The inserts show the current curve of the MEG device at different time slots.

Since water is the main energy source of MEG, it is conceived that the electricity regeneration can be realized by re-absorption from air after dehydration of MEG. Thus it is reasonable to define the recharging time as the time needed to dehydrate the water from MEG to enable next-time electric generation. Furthermore, the cyclic electric performance of the MEG was tested through water adsorption-dehydration-adsorption cycles. The resulting current curves are demonstrated in Fig. R3-9. The first absorption cycle enables MEG deliver a sustained current

of about 0.22 mA. After that, MEG undergoes dehydration process by drying at 20% RH and room temperature about 24 h. The subsequent absorption cycles also drive MEG to generate sustained current output despite the current outputs become slightly smaller, displaying decent cycle electric performance of the MEG.

Fig. R3-9 Cyclic electric performance of MEG with load resistance of 1 kΩ in water adsorption-desorption cycles under 70% RH and room temperature.

Additionally, long-time performance has been added in the supporting information on page 6 in Fig. S3 and discussed in the manuscript on page 7 as follows: “Beyond that, the current output gradually decays due to saturation of moisture absorption but persists for over one month (Fig. S3).” Furthermore, the cyclic electric performance has been added in the supporting information on page 13 in Fig. S12 and discussed in the manuscript on page 10 as follows: “In addition, a decent cyclic current output was also observed by moisture absorption-desorption process (Fig. S12). These results further indicate that moisture is the prime energy source of MEG.”

7. The author analyzed the effect of CaCl₂ on the output current of the MEG device. Will the CaCl₂ influence the sustained output of the MEG device? I want to see this point.

[Author’s reply]: Thanks a lot for your comments. As shown in Fig. R3-10, the incorporation of CaCl₂ brings in a rise of I_{sc} firstly while a certain decline beyond 3.8 wt% of CaCl₂. The optimized concentration is about 2.4 wt% CaCl₂ for MEG, which is used in this work. And the

results in Fig. 1d-e show that MEG with 2.4 wt% CaCl₂ generates a continuous DC V_{oc} of about 1.30 V for more than 90 hours and a sustained current output of ca. 0.25 mA for about 17 hours when connected to an external resistance of 1 k Ω (Fig. 1e).

Fig. R3-10 I_{sc} and V_{oc} of MEG units with different CaCl₂ concentration from 0 to 4.8 wt% under 80% RH.

To investigate the effect of CaCl₂ on the sustained output of the MEG device, MEGs with 1.2 wt% and 3.8 wt% CaCl₂ have been fabricated to test the long term V_{oc} and current with the loading resistance of 1 k Ω . Fig. R3-11 shows that MEG with 3.8 wt% CaCl₂ delivers a sustained V_{oc} of about 1.3 V for more than 90 hours and MEG with 1.2 wt% CaCl₂ also delivers a continuous V_{oc} of about 1.3 V for more than 90 hours. The CaCl₂ concentration shows little influence on the sustainability of V_{oc} . In addition, the loading currents of both MEG with 3.8 wt% and 1.2 wt% CaCl₂ keep steady for over 17 hours with value of about 100 μ A. Both the V_{oc} and loading current output show great sustainability over a long time. Therefore, the CaCl₂ exerts little influence on the sustained output of the MEG device.

Fig. R3-11 (a) The V_{oc} curves for MEGs with 1.2 wt% and 3.8 wt% CaCl_2 under open air and room temperature. (b) The current curves for MEGs with 1.2 wt% and 3.8 wt% CaCl_2 with the loading resistance of 1 k Ω under 70% RH and room temperature.

8. The authors use the PVA with a molecular weight (MW) of ~65,000. Will the performance of MEG be influenced by the MW of the PVA? I want to see some performance comparisons of MEG with different PVA molecules (e.g., PVA with larger MW).

[Author's reply]: Thanks a lot for your suggestions. To investigate the influence of M_w of PVA on the performance of MEG, PVA with larger M_w about 88000 has been selected to prepare MEG by the same procedure. The results of V_{oc} and I_{sc} have been shown in Fig. R3-12. The V_{oc} shows almost the same while the I_{sc} decreases from about 406 to 300 μA , mainly deriving from larger viscosity of PVA with larger M_w slowing down ion diffusion process.

Fig. R3-12 The V_{oc} and I_{sc} for MEG with different molecular weight of PVA (61000 and 88000, respectively).

9. Could the authors provide some SEM images of their PVA-alginate hydrogel in both the fully-dried state and water-absorbed states?

[Author's reply]: Thanks a lot for your suggestions. Fig. R3-13a and b demonstrate the morphology of hydrogel exposure at 80% RH for 24 hours. Fig. R3-13c and d show the SEM

image of hydrogel after drying at 20% RH for one week. There seems little difference for both of PVA-alginate hydrogels in the fully-dried state and water-absorbed state.

Fig. R3-13 SEM images of hydrogels with water-absorbed state (a, b) and fully-dried state (c, d).

10. For the 1.31 mA cm^{-2} MEG, the surface area is only 0.01 cm^2 . What is the thickness of this specific device? What is the accuracy of the surface area measurement? What is the long-term current performance of this device?

[Author's reply]: Thanks for your comments. The thickness of the MEG device is about 2 mm, which is same with other sizes of MEGs. Fig. R3-14a clearly shows the picture of a MEG device with 0.01 cm^2 . As shown in Fig. R3-14b, we use a spacer by laser cutting to fix the size of hydrogel when the hydrogel was casted on the bottom electrode. The accuracy of a spacer is decided by laser beam, which is about $\pm 0.05 \text{ mm}$. Thus the side length accuracy of MEG device with $1 \times 1 \text{ mm}^2$ is about $\pm 0.05 \text{ mm}$. In addition, Fig. R3-14c shows that the MEG unit with 0.01 cm^2 can generate a short-circuit current (I_{sc}) of about $13 \text{ } \mu\text{A}$ over 8000s, indicating its continuous current output for long-term use.

Fig. R3-14 (a) The picture of a MEG unit with 0.01 cm^2 size. (b) The spacer to decide the device area of MEG. (c) The long-term I_{sc} curve of 0.01 cm^2 MEG.

11. A plot of current density vs device area should also be provided in addition to Fig. 2f.

[Author's reply]: Many thanks for your suggestions. The I_{sc} curves with the time vs device area have been shown in Fig. R3-15 and Fig. S10a in the supporting information on page 12.

The manuscript on page 9-10 have been modified: "Excitingly, I_{sc} demonstrates two orders of magnitude upgrade (ca. 2.14 mA) when the size increases to 9 cm^2 . The corresponding I_{sc} curves for different sizes have been plotted in Fig. S10. Both MEGs with sizes of 4 and 9 cm^2 show the large DC current output of more than 1 mA even beyond 4000s."

Fig. R3-15 The I_{sc} curves with the time versus device area.

12. Atoms in the ESP distribution of the PVA, AlgNa, AlgCa in Fig. 1b should be labeled.

[Author's reply]: Thanks a lot for your comments. The atoms have been labeled in Fig. R3-16 as following and Fig. 1b in the manuscript.

Fig. R3-16 Device structure of MEG and corresponding electric output. (a) Schematic illustration of a single MEG unit. The right scheme shows that active material consists of supramolecular AlgNa/Ca network within PVA hydrogel, which is formed through replacing partial Na^+ ions by Ca^{2+} ions as the egg-box crosslinked points. (b) The ESP distribution of PVA, AlgNa, AlgCa from left to right by DFT calculations. The units of color bar are in kcal mol^{-1} . Surface local minima and maxima of ESP are represented as orange and blue spheres, respectively. (c) Surface area and corresponding area percent in each ESP range on the vdW surface of PVA, AlgNa, AlgCa from left to right correspondingly. (d) The V_{oc} (red curve) of an MEG device over time under an open environment with fluctuating RH. The ambient RH

(black curve) was synchronously recorded. (e) The current output of an MEG device with 1 k Ω external resistor under the evolution of time at 70% RH. The insert shows the circuit diagram of MEG with one resistor. The area of one single unit is fixed at 1 cm² and the test temperature is about 22 °C, unless otherwise stated.

13. What is the freezing point of the PVA-alginate hydrogel? And what is the state of the PVA-alginate hydrogel at -25 °C? Pure PVA should freeze at -20 °C.

[Author's reply]: Thanks a lot for your comments. We have tried to measure the freezing point of hydrogel by DSC. Fig. R3-17a shows that there is no freezing point despite the lowest temperature down to -70 °C. That means our hydrogel shows great anti-freezing property because CaCl₂ effectively lowers the freezing point of water [Journal of Materials Chemistry A. 2020; 8: 25390]. Besides, PVA-AlgNa hydrogel can be still twisted at -26.6 °C as shown in Fig. R3-17b, suggesting a superior elasticity of PVA-AlgNa hydrogel at low temperature.

Fig. R3-17 (a) The DSC curve of PVA-AlgNa hydrogel heating from -70 to 60 °C with a heat rate of 2 °C min⁻¹. (b) Image of twisting state of hydrogels placed at -26.6 °C. Scale bar: 1 cm.

14. The color scale of Fig. S7 should be provided at least in A.U.

[Author's reply]: Many thanks for your suggestions. The color scale has been provided in A.U as shown in Fig. R3-18 as following and Fig. S17 in the supporting information. The similar change is also made in Fig. S18.

Fig. R3-18 2D Raman mapping. (a) The moisture capturing by top surface of PVA-AlgNa based hydrogel after exposing in air for 45 min. (b) The moisture capturing by bottom surface of PVA-AlgNa based hydrogel with the same exposure time. (c) The moisture capturing by top surface of PVA after exposed in air for 45 min.

15. Fig. 4d is not described and explained in the main text.

[Author’s reply]: Many thanks for your comments. Fig. 4d shows O-H stretching band ($\nu(-OH)$) gradually becomes stronger with the time revolution in time-resolved 1D FTIR, which can be decoupled to get 2D-FTIR spectroscopy. Therefore, we can gain an insight into deep investigation of dynamic water and ion diffusion process.

The detailed description is shown in the manuscript on page 14: “On this basis, we further uncover the underpinning mechanism of moisture-initiated electric generation through deep investigation of dynamic water and ion diffusion process. 1D FTIR spectroscopy in Fig. 4d shows the strength of $\nu(-OH)$ gradually increases accompanied with the absorption of moisture. Based on this time-resolved 1D FTIR, 2D-FTIR spectroscopy is got by decoupling overlapped bands of O-H stretching band ($\nu(-OH)$), which reveals water diffusion and water gradient construction from the molecular level.”

16. The current per unit area for the large-scale MEG array should also be provided. (OK)

[Author’s reply]: Many thanks for your suggestions. Fig. R3-19a shows that I_{sc} enlarges linearly with the parallel number of MEG units increasing from 10 to 280 units. Furthermore,

the average current density (J_{sc}) is calculated to be about 0.23 mA cm^{-2} for each MEG unit in Fig. R3-19b.

Fig. R3-19 (a) The plot of I_{sc} related to the parallel number of MEG units. (b) The current density for the large-scale MEG array.

The plot of J_{sc} versus parallel number has been added in the supporting information of Fig. S27 on page 21. And the manuscript on page 16 has been modified as following: “What’s thrilling is that the current is scaled up to ca. 65 mA with only 280 parallel units for the first time, which is one or two orders of magnitude better than previous integrated MEGs.^{35, 43, 44} The average current density output is about 0.23 mA cm^{-2} for each MEG unit (Fig. S27).”

17. Can the authors elaborate on the process by which the PVA-alginate solution underwent gelation? Was the freeze-thaw method used or did the solution gelate at ambient temperature?

[Author’s reply]: Many thanks for your comments. In our work, the PVA-AlgNa solution gels at ambient environment of $25 \text{ }^{\circ}\text{C}$ and 65% RH for over 24 hours as shown in Fig. R3-20b. Before gelation, PVA-AlgNa solution is flowable when tilting glass bottle (Fig. R3-20a). In clear contrast, the PVA-AlgNa solution is able to in situ form hydrogel, indicating that the solution can be gelated at ambient temperature.

Fig. R3-20 (a) PVA-AlgNa solution before gelation. (B) PVA-AlgNa hydrogel after gelation.

18. Can the authors confirm if the asymmetric moisture absorption of the MEG is solely due to the asymmetric electrode design, or are there additional strategies implemented to promote this asymmetric moisture absorption?

[Author's reply]: Many thanks for your comments. In general, unidirectional moisture stimuli and asymmetric moisture absorption are two strategies to construct water gradient of MEG with homogeneous active materials. [ACS Materials Letters. 2021; 3: 193] In our work, the asymmetric moisture absorption of the MEG mainly benefits from the asymmetric electrode design. The top electrode with holes provides prior chance to capture moisture while the bottom substrate obstructs the moisture, spontaneously forming water gradient within the material.

REVIEWER COMMENTS

Reviewer #1 (Remarks to the Author):

The authors have done good job incorporating reviewer feedback to make this manuscript much stronger than it was as submitted earlier. Overall, the range of the claims is appropriate for the evidence provided.

Reviewer #2 (Remarks to the Author):

I have the following comments:

- 1、 Figure R2-10 shows that the MEGs using pure PVA hydrogel can maintain an output performance of 0.8 V and 0.14 μ A. However, it is known that most PVA lacks humidity-driven power generation performance, due to its limited oxygen-containing functionality. Please explain the main reason and influencing factors of this high performance of MEGs' using PVA.
- 2、 Can the authors provide more information about the impact of crosslinker concentration on the hydrogel structure and performance of MEGs? Is the presence of porous structure a critical factor in the power generation process ?
- 3、 In Figure R3-2, a reversed $c(\text{Ca})$ from negative into positive is observed. Similar results were also observed in Na and Cl ions. Does the transport of Cl ions (compared to Ca ions) result in a reversal of the electrical output?
- 4、 Can this device spontaneously generate electricity using the same Au electrodes after a prolonged discharge period, as shown in the Ref (Lan L, Chen S, Liu X, Xiong J, Ping J, Ying Y. The role of electrochemistry and surface chemistry in water-induced electricity generation from electrically conductive solids. ChemRxiv. Cambridge: Cambridge Open Engage; 2022). Additionally, if the upper electrode remains unperforated, can the device produce electricity without a moisture gradient?

Reviewer #3 (Remarks to the Author):

The authors have addressed most of my comments carefully. I appreciate their efforts. I have some additional questions regarding the response. I think the manuscript can be accepted after the authors address my further questions below,

1. Regarding my first question, in Figure R3-2, it seems that the Na ion of the bottom and top surfaces has a less difference compared to Ca and Cl ions (one order of magnitude). This observation is consistent with the nearly same Na concentration difference measured after 5 hours and 15 days. Does this indicate that the Na ions are less or not involved in the current flow when the hydrogel is in short circuit condition? Does this observation reflect that Ca and Cl ions dominate the current flow in the hydrogel in the short circuit condition? Considering that the Ca ions are divalent cations with strong interaction with the alginate polymers, should the Ca ion dissociation be more difficult than Na ions (The scheme in Fig. 1a shows free-moving Na ions and associated Ca ions with the alginate)? This seems to contradict the observed ion difference on the top and bottom surfaces of the hydrogel. I think this point should be further elaborated. I would suggest the authors perform additional experiments using alginate acid as the component during synthesis instead of sodium alginate.
2. I have additional questions on the long-term performance of the MEG devices. It seems that the open-circuit voltage is more stable than the short-circuit current. Can the author justify the difference and its rationale?
3. Regarding my question 5, I think the author cannot simply multiply short-circuit current and open-circuit voltage to calculate the electrical output because they cannot be obtained simultaneously. I would suggest using I^2R with R representing the external load.
4. Following my question 8, the authors observed that with higher PVA Mw, the current is decreased because of the slowed ion diffusion process. What is the sustained performance of the supramolecular hydrogel using PVA with higher Mw? Will this slower ion diffusion process lead to more sustained current output?

5. An additional question from my side is: will the thickness of the device influence the performance? In the manuscript, it is mentioned that a typical thickness is 2 mm. It is known sample thickness could affect the diffusion of water.

We are very grateful for the time and effort made by the Reviewers, especially for their helpful suggestions and constructive comments. We have revised the manuscript and supporting information accordingly. The changes are highlighted in color. The detailed responses are given as follows:

RESPONSE TO REVIEWERS' COMMENTS

Reviewer #1 (Remarks to the Author):

The authors have done good job incorporating reviewer feedback to make this manuscript much stronger than it was as submitted earlier. Overall, the range of the claims is appropriate for the evidence provided.

[Author's reply]: Thanks a lot for your encouraging comments.

Reviewer #2 (Remarks to the Author):

I have the following comments:

1. Figure R2-10 shows that the MEGs using pure PVA hydrogel can maintain an output performance of 0.8 V and 0.14 μ A. However, it is known that most PVA lacks humidity-driven power generation performance, due to its limited oxygen-containing functionality. Please explain the main reason and influencing factors of this high performance of MEGs' using PVA.

[Author's reply]: Many thanks for your comments. It is well known that pure PVA features rich hydroxyl functional groups and decent moisture absorption. [Advanced Functional Materials, 2021, 31: 2009172.] That means PVA has great potential to be used as active material for MEGs and protons can be dissociated from hydroxyl groups of PVA chains despite more dissociation energy is needed due to strong internal hydrogen bonds. In fact, previous work has reported that PVA film-based MEG delivered a voltage of about 0.4 V, demonstrating moisture-electric power capability of PVA. [Energy & Environmental Science, 2019, 12: 972.] In this work, MEGs using pure PVA hydrogel can maintain an output performance of ca. 0.8 V and 0.14 μ A. The larger voltage output here is mainly attributed to different porous networks

between PVA film and PVA hydrogel. Normally, PVA hydrogel owns looser polymeric network and porous structure compared to PVA film, which is beneficial to the movement of dissociated ions. [Chemistry-A European Journal, 2013, 19: 7118.] Thus, a better electricity is observed in PVA hydrogel-based MEG in our work. Except for porous structure, other influencing factors may include molecular weight, degree of hydrolysis, etc. In the next work, we plan to systematically investigate the influencing factors of MEG based on PVA. We hope the systematic results may be got soon.

2. Can the authors provide more information about the impact of crosslinker concentration on the hydrogel structure and performance of MEGs? Is the presence of porous structure a critical factor in the power generation process?

[Author's reply]: Thanks a lot for your comments. As shown in Fig. R2-1, the incorporation of CaCl_2 brings in a rise of I_{sc} firstly while a certain decline beyond 3.8 wt% of CaCl_2 . The optimized concentration is about 2.4 wt% CaCl_2 for MEG. By adding crosslinker of CaCl_2 , supramolecular AlgCa/Na ionically cross-linked network is formed with abundant carboxyl functional groups (e.g., $-\text{COONa}$ and $-\text{COOCa}$). Thanks to the increased moisture capturing and dissociable ion diffusion within supramolecular network, the incorporation of CaCl_2 at appropriate amount contributes to a better I_{sc} (Fig. R2-1). However, the excessive CaCl_2 will exert little impact on the crosslinking degree of hydrogel structure since the full crosslinking Ca^{2+} with $-\text{COO}^-$ of AlgNa chains has been achieved. Moreover, the excessive CaCl_2 may lead to attenuated water gradient by absorbing excess water, thus lowering power output.

Fig. R2-1 I_{sc} and V_{oc} of MEG units with different CaCl_2 concentration from 0 to 4.8 wt% under 80% RH.

As for the role of porous structure, it is well considered that designing porous structures with higher specific area is beneficial to increase the accessibility of water, as well as charged ions. [Adv. Mater. 2020, 32, 2003722; Energy Environ. Sci. 2016, 9, 912.] Moreover, as we discussed above, in contrast with PVA film, PVA hydrogel owns loose polymeric network and porous structure, which benefits the absorption of water molecules and following diffusion of ions. As a result, a better electricity output is observed in PVA hydrogel-based MEG in our work.

The detailed discussion has been stated in the manuscript on page 10: “The increased I_{sc} stems from more moisture capturing and conducive ion diffusion within supramolecular network. However, the excessive CaCl_2 may lead to attenuated water gradient by absorbing excess water, thus lowering power output.”

3. In Figure R3-2, a reversed $c(\text{Ca})$ from negative into positive is observed. Similar results were also observed in Na and Cl ions. Does the transport of Cl ions (compared to Ca ions) result in a reversal of the electrical output?

[Author’s reply]: Thanks a lot for your comments. As shown in Fig. R2-2, the transports of Ca, Na, and Cl ions have the same tendency during the charge-discharge process. Without moisture stimulation, free ions like Na and Cl ions are uniformly dispersed in hydrogels with random Brownian motions. This kind of random motions make little contribution to the electrical output. With moisture stimulation, Ca ions can be released from the polymer chains (Alg chains). Then the charge separation between Ca and Alg chains as well as directional ion diffusion account for the electric generation. [Energy & Environmental Science, 2019, 12: 972] That is to say, the charge separation between Ca and Alg chains generates potential difference, which serves as the driving force to facilitate other ions like Cl ions to migrate directionally. Since water induces Ca ions diffusion from top to bottom, the transported ions tend to attract

the counterions by the electrostatic attraction, leading to the potential neutralization. [J. Am. Chem. Soc. 2021, 143, 14242] Therefore, the transport of Cl ions tends to re-unite the free Ca and Na ions, which results in a neutralization instead of reversal of the electrical output.

Fig. R2-2 The chemical component characterization of detective ion variation between the top and bottom surface of MEG in different states by EDS. (a) Ca element. (b) Cl element. (c) Na element.

4. Can this device spontaneously generate electricity using the same Au electrodes after a prolonged discharge period, as shown in the Ref (Lan L, Chen S, Liu X, Xiong J, Ping J, Ying Y. The role of electrochemistry and surface chemistry in water-induced electricity generation from electrically conductive solids. ChemRxiv. Cambridge: Cambridge Open Engage; 2022). Additionally, if the upper electrode remains unperforated, can the device produce electricity without a moisture gradient?

[Author's reply]: Thanks a lot for your pertinent comments. Firstly, Au-Au electrode couples were used to prepare MEGs and the upper electrodes were perforated. After the short-circuit discharge treatment, the voltage of as-prepared MEG was tested as shown in Fig. R2-3. The

MEG gradually re-generates a voltage of about 0.47 V with self-charging of about 5 h.

Fig. R2-3 The voltage curve of a MEG unit with Au-Au electrode couple after a prolonged discharge period of about 5 h at 80% RH and room temperature.

Secondly, MEGs with Au-Au electrode couples were prepared and the upper electrodes remained unperforated. As a result, MEG without pores delivers a very low voltage of about 1.55 mV, which is much lower than that of MEG with pores (Fig. R2-4). It indicates that the moisture absorption and remained water gradient is important for the electric generation.

Fig. R2-4 The electric comparison of MEGs by using Au-Au electrode couples with and without pores at 80% RH and room temperature.

Reviewer #3 (Remarks to the Author):

The authors have addressed most of my comments carefully. I appreciate their efforts. I have some additional questions regarding the response. I think the manuscript can be accepted after the authors address my further questions below,

[Author's reply]: Thanks a lot for your encouraging comments.

1. Regarding my first question, in Figure R3-2, it seems that the Na ion of the bottom and top surfaces has a less difference compared to Ca and Cl ions (one order of magnitude). This observation is consistent with the nearly same Na concentration difference measured after 5 hours and 15 days. Does this indicate that the Na ions are less or not involved in the current flow when the hydrogel is in short circuit condition? Does this observation reflect that Ca and Cl ions dominate the current flow in the hydrogel in the short circuit condition? Considering that the Ca ions are divalent cations with strong interaction with the alginate polymers, should the Ca ion dissociation be more difficult than Na ions (The scheme in Fig. 1a shows free-moving Na ions and associated Ca ions with the alginate)? This seems to contradict the observed ion difference on the top and bottom surfaces of the hydrogel. I think this point should be further elaborated. I would suggest the authors perform additional experiments using alginate acid as the component during synthesis instead of sodium alginate.

[Author's reply]: Thanks a lot for your pertinent comments and suggestions. To figure out the role of Na ions on the current output, pure PVA-alginate acid hydrogel and pure PVA-AlgNa hydrogel without CaCl₂ have been fabricated to prepare MEGs. The electric outputs are depicted in Fig. R3-1. MEG using Alginate acid shows a lower current of about 9.4 μA compared to MEGs using AlgNa (ca. 77.6 μA), which suggests the dissociated Na ions make positive contributions to the current output of MEGs.

Fig. R3-1 The V_{oc} and I_{sc} of MEGs using Alginate acid compared to that of MEGs using AlgNa at 80% RH and room temperature.

MEGs rely on the interaction of active materials with moisture. The improved water capturing capability can empower MEG with sufficient chemical conversion energy to trigger ample ions diffusion, consequently inducing high power output. Fig. R3-2 shows that the absorption energy of Alginate acid-H₂O is about -63.31 kJ/mol by DFT calculation, which is smaller than that of AlgNa-H₂O (-106.16 kJ/mol). The weaker interaction between Alginate acid and H₂O leads to inferior chemical energy conversion. Therefore, less protons are dissociated from Alg chains and MEG using Alginate acid generates inferior current. In clear contrast, the absorption energy (-142.94 kJ/mol) of AlgCa-H₂O is the strongest, enabling sufficient chemical conversion energy to trigger Ca ions to dissociate from the polymer chains and thus rich Ca ions transport is observed. As a positive result, MEGs based on AlgNa/Ca hybrid hydrogels output large current. The electric output is well consistent with the water absorption energy of hybrid hydrogels. It further verifies the moisture is key to the electric generation for MEGs.

Fig. R3-2 Interaction region indicator maps and the corresponding interaction energies (kJ/mol) of AlgCa-H₂O, AlgNa-H₂O, Alginate acid-H₂O. The $\text{sign}(\lambda_2)\rho$ is mapped on the isosurfaces.

The Ca ions diffusion was also discussed in the manuscript on page 15: “It is worth noting that the absolute value of $\Delta c(\text{Ca})$ becomes larger, which suggests that moisture enables extra dissociated Ca^{2+} ions release from polymer chains (-COOCa) and then triggers these Ca^{2+} ions migration. Whereafter, the MEG is discharged through short-circuit treatment. A reversed $\Delta c(\text{Ca})$ from negative into positive is observed, indicating an opposite transport process. Apparently, the moisture-triggered ions migration directly contributes to the outstanding electric generation of MEG, coupling with the joint contribution from directional migration of Cl^- , Na^+ (Fig. S21).” Besides, Interaction region indicator of Alginate acid-H₂O has been added and described in Fig. S21c of supporting information on page 19 and discussed in the manuscript of page 13: “The weak interaction is also observed between Alginate acid and H₂O (-63.31 kJ/mol) (Fig. S21c).”

2. I have additional questions on the long-term performance of the MEG devices. It seems that the open-circuit voltage is more stable than the short-circuit current. Can the author justify the difference and its rationale?

[Author’s reply]: Thanks a lot for your comments. As we have discussed in the manuscript: “A sustained current lasts for more than 120 h accompanied by the concurrent adsorption of water molecules, which couples with the ion dissociation process in the MEG even after reaching absorption saturation state (Fig. S14). It well demonstrates that moisture absorption process is directly related to electricity generation of MEG.” The moisture absorption process is very fast firstly and then it becomes slow after reaching absorption saturation state (Fig. R3-3). That means the chemical conversion energy of moisture-to-water is large firstly and becomes smaller under saturation state. The large absorption energy enables to trigger more ions dissociation, leading to larger current density. On the contrary, the current of MEG becomes smaller quickly after reaching saturation state due to inferior absorption energy conversion.

Fig. R3-3 Moisture uptake capability and current output of MEG versus time synchronously. (a) The mass increase of MEG by moisture adsorption versus time under 80% RH and room temperature. (b) The concurrent current output of MEG versus time at the same condition. (c-e) The tested current output of MEG at different time slot.

As we know, the voltage output is closely related to water gradient. In Fig. R3-4, it clearly demonstrates the long existence of a built-in water gradient within the hydrogel after one week, which accounts for long-term stability of voltage for MEG device (more than 90 h). [Nature, 2020, 578: 550.] The long-standing water gradient mainly stems from strong water adsorption energy of PVA-AlgNa based supramolecular hydrogel with asymmetrical absorption as indicated by DFT calculations in our manuscript. From above discussion, it is reasonable that the voltage remains stable over a long time and current becomes small quickly after saturation.

Fig. R3-4 Cross-sectional 2D Raman mapping of PVA-AlgNa based hydrogel with the depth after one week.

3. Regarding my question 5, I think the author cannot simply multiply short-circuit current and open-circuit voltage to calculate the electrical output because they cannot be obtained simultaneously. I would suggest using I^2R with R representing the external load.

[Author's reply]: Thanks a lot for your useful suggestions. The load current of MEG device with the external resistor of $1\text{ k}\Omega$ was implemented to calculate the power output of MEG with time as shown in Fig. R3-5. The result shows the similar trend of power output of MEG as discussed before.

Fig. R3-5 (a) The current output of MEG versus time under 80% RH and room temperature. (b) The calculated power output of MEG versus time at the same condition.

4. Following my question 8, the authors observed that with higher PVA M_w , the current is decreased because of the slowed ion diffusion process. What is the sustained performance of the supramolecular hydrogel using PVA with higher M_w ? Will this slower ion diffusion process lead to more sustained current output?

[Author's reply]: Many thanks for your comments. Based on our last result, the V_{oc} shows almost the same while the I_{sc} decreases from about 406 to $300\ \mu\text{A}$ for PVA with higher M_w of 88k (Fig. R3-6). It mainly derives from larger viscosity of PVA with higher M_w , which slows down ion diffusion process.

Fig. R3-6 The V_{oc} and I_{sc} for MEG with different molecular weight of PVA (61k and 88k, respectively) at 80% RH and room temperature.

To further investigate the influence of molecular weight (M_w) of PVA on sustainability of MEGs, the current outputs of both MEGs with PVA-61k and PVA-88k were tested coupling with a load resistor of 1 k Ω for an extended time. Fig. R3-7 clearly shows that both MEGs with PVA-61k and PVA-88k generate sustained current outputs even over 100000 s. That is to say, MEG using PVA with higher M_w exerts little influence on its sustainability at the same timescale. It is mainly attributed to that the water absorption capability remains the same for both MEGs with higher and lower M_w of PVA. The water absorption capability of MEGs is predominated by AlgNa/Ca in our case. As we discussed above, moisture absorption process is directly related to electricity generation of MEG. Therefore, the M_w of PVA exerts little impact on the water absorption process of MEG from the air but limits the ion transport due to larger viscosity of PVA-88k, leading to a smaller current but great sustainability.

Fig. R3-7 The current outputs of MEG devices based on PVA-61k (a) and PVA-88k (b) with 1

k Ω external resistor under the evolution of time at 70% RH and room temperature, respectively.

5. An additional question from my side is: will the thickness of the device influence the performance? In the manuscript, it is mentioned that a typical thickness is 2 mm. It is known sample thickness could affect the diffusion of water.

[Author's reply]: Thanks very much for your comments. To gain a deep insight into the influence of thickness, the MEGs with different thickness were supplemented to measure the voltage and current output. As shown in Fig R3-8, the MEG device delivers stable voltage output of ca. 1.25 V, while the current density increases first and then decreases with increasing thickness. Thereby, herein the thickness of the device is chosen as 2 mm.

Fig. R3-8 The V_{oc} and J_{sc} plotted against film thickness for a device size of 0.25 cm². The test condition is at 80% RH and room temperature.

As we know, the voltage output is closely related to water gradient. The same RH condition provides the similar water gradient between upper and bottom surfaces of hydrogel no matter of the thickness. Consequently, the voltage maintains almost the same at different thickness. On the other hand, the current density mainly depends on the dissociable ions and ion diffusion rate at a suitable thickness range and the same RH. The more dissociable ions and faster ion diffusion rate, the higher current density. Ion diffusion rate is about the same under the similar polymer network. Meanwhile, the hydrogel with larger thickness contains more dissociable ions, which increases the number of mobile ions under high RH stimulation at a suitable

thickness range. Thus, the current density gradually increases with the thickness firstly. However, when the hydrogel is too thick, the too long ion diffusion path primarily hinders the transport of ions and reduces the ion diffusion rate, lowering the current density gradually followed by the peak value.

The detailed discussion has been added in the supporting information on page 12 of Fig. S10c. The manuscript on page 10 was revised to: “Similar phenomenon is observed for MEGs with different thickness. MEG with 2.5 mm thickness shows decreased current density due to long ion diffusion path (Fig S10c). Here, the samples with AlgNa concentration of 1 wt%, CaCl₂ concentration of 2.4 wt% and 2 mm thickness are used in this work unless otherwise stated.”

REVIEWERS' COMMENTS

Reviewer #2 (Remarks to the Author):

All the questions have been well answered. I suggest to accept this manuscript.

Reviewer #3 (Remarks to the Author):

The authors solved my concerns appropriately. I suggest its publication in nature communication.

RESPONSE TO REVIEWERS' COMMENTS

Reviewer #2 (Remarks to the Author):

All the questions have been well answered. I suggest to accept this manuscript.

[Author's reply]: Thanks a lot for your encouraging comments.

Reviewer #3 (Remarks to the Author):

The authors solved my concerns appropriately. I suggest its publication in nature communication.

[Author's reply]: Thanks a lot for your encouraging comments.